# Blood biomarkers of Alzheimer's disease and progression across different stages of cognitive decline in the community

Martina Valletta [1] ✉, Davide Liborio Vetrano [1,2], Caterina Gregorio [1], Debora Rizzuto[1,2], Bengt Winblad [3,4], Marco Canevelli[1,5,6], Sarah Andersson[7], Matilda Dale [7], Claudia Fredolini [7], Erika J. Laukka[1,2], Laura Fratiglioni[1] & Giulia Grande [1,2]

Blood biomarkers of Alzheimer's disease (AD) are promising for dementia prediction, but their association with progression across intermediate stages of cognitive decline in the general population remains unclear. We followed 2148 dementia-free individuals from a Swedish population-based cohort for up to 16 years. Associations between baseline AD blood biomarkers and transitions between normal cognition, mild cognitive impairment (MCI), and dementia were examined. Lower amyloid-β42/40 ratio and higher phosphorylated-tau181 (p-tau181), p-tau217, total-tau, neurofilament light chain (NfL), and glial fibrillary acidic protein (GFAP) were associated with faster progression from MCI to all-cause and AD dementia, with the strongest associations for NfL and p-tau217. Elevated NfL and GFAP were linked to reduced MCI reversion to normal cognition, whereas no biomarker was associated with MCI development from normal cognition. These findings show robust group-level associations and indicate that AD blood biomarkers may help stratify dementia risk at the MCI stage in the community.

Blood biomarkers of Alzheimer's disease (AD) have emerged as reliable indicators of AD pathology and accurate predictors of future clinical AD in specialized clinical settings[1,2]. Compared to cerebrospinal fluid (CSF) and positron emission tomography (PET), blood biomarkers are cost-effective and minimally invasive, making them promising candidates for large-scale applications[3,4]. A growing body of studies has started to support the extension of these findings to the general population[5–8]. In line with this evidence, we recently demonstrated that phosphorylated tau (p-tau) isoforms, neurofilament light chain (NfL), and glial fibrillary acidic protein (GFAP) can accurately predict future all-cause and AD dementia in a community-based cohort[9]. However,

evidence regarding their association with progression across intermediate stages of cognitive decline remains limited.

Dementia is the end stage of a prolonged preclinical and prodromal phase, during which neuropathology accumulates and mild cognitive deficits appear[10,11]. This early stage holds clinical significance as therapeutic interventions are expected to yield the greatest benefits before dementia is clinically apparent[12]. While a few clinical studies have shown that individuals with mild cognitive impairment (MCI) and elevated levels of AD blood biomarkers are at high risk of progression to dementia[13–17], data from population-based studies are still lacking. Moreover, how AD blood biomarkers influence the transitions across

[1]Aging Research Center, Department of Neurobiology, Care Sciences and Society, Karolinska Institutet and Stockholm University, Stockholm, Sweden. [2]Stockholm Gerontology Research Center, Stockholm, Sweden. [3]Division of Neurogeriatrics, Department of Neurobiology, Care Sciences and Society, Karolinska Institutet, Solna, Sweden. [4]Theme Inflammation and Aging, Karolinska University Hospital, Huddinge, Sweden. [5]Department of Human Neuroscience, Sapienza University, Rome, Italy. [6]National Centre for Disease Prevention and Health Promotion, Italian National Institute of Health, Rome, Italy. [7]Affinity Proteomics Stockholm, Science for Life Laboratory, Department of Protein Science, School of Engineering Sciences in Chemistry, Biotechnology and Health (CBH), Royal Institute of Technology (KTH), Solna, Sweden. ✉e-mail: martina.valletta@ki.se

different stages of cognitive decline, including the development of MCI, the reversion to normal cognition and the progression to dementia, remains poorly understood. Addressing those questions could enhance early identification of at-risk cognitive phenotypes and improve preventive and therapeutic strategies.

We aimed to investigate the association between the levels of six AD blood biomarkers—both individually and in combination—and the progression across different stages of cognitive decline (i.e., normal cognition, MCI, and dementia) in a population-based cohort of over 2000 older adults followed for up to 16 years.

## Results
At baseline, median age of study participants was 72.2 years, 61.5% were females, and 35.4% had a university level of education or higher. Table 1 summarizes the characteristics of study participants. At baseline, 381 (17.7%) participants had MCI. During a mean follow up of 9.6 (4.3) years, 286 (16.2% of participants who did not have MCI at baseline) participants developed MCI and 364 (16.9%) developed all-cause dementia, which in 212 cases (58.2%) was of AD type; 1101 (51.3%) participants died.

### Individual blood biomarkers of AD and HR for progression across different cognitive stages
Figure 1 and Supplementary Fig S1 show the HR for the transitions across different cognitive stages according to baseline AD blood biomarkers levels (i.e., continuous values). Elevated levels of all

biomarkers, except for the Aβ42/40 ratio, were associated with a faster progression from MCI to all-cause and AD dementia, following a non-linear relationship. Elevated levels of GFAP were also associated with lower hazard of reversion from MCI to normal cognition. No association was found between AD blood biomarkers and the progression from normal cognition to MCI.

Similar results were obtained when baseline AD blood biomarkers were dichotomized using predefined cut-off values, as shown in Table 2 and Supplementary Table S1. In basic models, a faster progression from MCI to all-cause and AD dementia was observed in participants with high, compared to low, levels of p-tau181, p-tau217, t-tau, NfL, and GFAP. A lower Aβ42/40 ratio was also associated with faster progression from MCI to all-cause and AD dementia. The strongest associations with all-cause and AD dementia were shown by NfL (HR 1.84, 95% CI 1.43, 2.36 for all-cause and HR 2.34, 95% CI 1.77, 3.11 for AD dementia) and p-tau217 (HR 1.74, 95% CI 1.38, 2.19 for all-cause and HR 2.11, 95% CI 1.61, 2.76 for AD dementia), followed by GFAP. Participants with high, compared to low, p-tau181, NfL, and GFAP also exhibited lower hazard of reversion from MCI to normal cognition. No association was found with the progression from normal cognition to MCI.

After further adjustment for chronic diseases (Table 2), the association between p-tau181 and MCI reversion was attenuated becoming non-significant, yet with similar point estimates.

### Subgroup and sensitivity analyses
The associations between biomarkers and progression from MCI to all-cause and AD dementia were slightly stronger in participants under 78 years old, compared to those older. Results for MCI reversion were instead mixed (Supplementary Tables S2 and S3). Sex-stratified analyses (Supplementary Tables S4 and S5) showed stronger associations between AD blood biomarkers and progression from MCI to all-cause dementia in women.

Most results remained consistent after excluding participants with MCI at baseline. Only the associations between p-tau181 and progression from MCI to all-cause and AD dementia and between NfL and GFAP and MCI reversion were attenuated, becoming non-significant (Supplementary Tables S6 and S7), although point estimates remained similar.

Sensitivity analyses using inverse probability weighting (IPW) to account for attrition yielded estimates that were consistent with the main results (Supplementary Tables S8 and S9). When applying cognitive impairment no dementia (CIND) operationalization instead of MCI, the results were comparable to those obtained with the MCI definition (data available upon request).

### Combinations of blood biomarkers of AD and HR for progression across different cognitive stages
Since p-tau217, NfL, and GFAP showed the strongest associations with our outcomes, we also explored the HR of transition across cognitive stages in relation to combinations of these biomarkers (Fig. 2 and Supplementary Tables S10–S13). The hazard of progression from MCI to all-cause and AD dementia increased with the number of elevated biomarkers at baseline. Participants with three elevated biomarkers had more than twice the hazard of progressing to all-cause dementia (HR 2.22, 95% CI 1.50, 3.28) compared to those with none, and an even higher hazard of progression to AD dementia (HR 3.71, 95% CI 2.22, 6.20) (Fig. 2 and Supplementary Tables S10 and S11). Considering pairs of biomarkers, the fastest progression from MCI to all-cause and AD dementia was observed in individuals with elevated levels of both p-tau217 and NfL (Supplementary Tables S12 and S13). Specifically, considering individuals with low levels of both biomarkers as the reference, the HRs of progression to all-cause dementia were 1.59 (95% CI 1.13, 2.22) for those with high NfL only, 1.71 (95% CI 1.11, 2.63) for those with high p-tau217 only, and 2.29 (95% CI 1.62, 3.24) for those

## Table 1 | Baseline characteristics of participants with available follow-up data

| | Overall N = 2148 | Age < 78 N = 1283 | Age ≥78 N = 865 |
|---|---|---|---|
| **Demographics** | | | |
| Age | 72.2 (60.9–81.2) | 66.1 (60.4–72.1) | 81.7 (78.4–87.8) |
| Sex (F) | 1322 (61.5%) | 723 (56.4%) | 599 (69.2%) |
| Education (University) | 760 (35.4%) | 590 (46.0%) | 170 (19.7%) |
| **MMSE** | 29.0 (28.0–30.0) | 29.0 (29.0–30.0) | 28.0 (27.0–29.0) |
| **Chronic diseases** | | | |
| Number of diseases | 3.0 (2.0–5.0) | 3.0 (2.0–4.0) | 5.0 (3.0–7.0) |
| Chronic kidney disease | 718 (33.4%) | 177 (13.8%) | 541 (62.5%) |
| Heart disease | 483 (22.5%) | 156 (12.2%) | 327 (37.8%) |
| Anemia | 236 (11.0%) | 57 (4.4%) | 179 (20.7%) |
| Cerebrovascular disease | 129 (6.0%) | 38 (3.0%) | 91 (10.5%) |
| Obesity | 276 (12.8%) | 198 (15.4%) | 78 (9.0%) |
| **Biomarkers** | | | |
| Aβ42/40 | 0.06 (0.05–0.07) | 0.06 (0.05–0.07) | 0.05 (0.05–0.06) |
| p-tau181 (pg/ml) | 1.2 (0.8–1.8) | 0.9 (0.6–1.4) | 1.7 (1.2–2.5) |
| p-tau217 (pg/ml) | 0.10 (0.06–0.18) | 0.07 (0.04–0.12) | 0.17 (0.11–0.29) |
| t-tau (pg/ml) | 0.8 (0.5–1.2) | 0.8 (0.5–1.1) | 1.0 (0.7–1.4) |
| NfL (pg/ml) | 18.3 (12.8–29.3) | 14.0 (10.8–18.3) | 30.4 (22.5–43.8) |
| GFAP (pg/ml) | 124.4 (81.9–193.8) | 96.0 (66.6–132.9) | 189.2 (136.2–284.0) |

Data are reported as median (Q1-Q3) for continuous variables and as *n* (%) for categorical variables. Missing data: 1 in education, 4 in MMSE.

Aß amyloid-beta, *GFAP* glial fibrillary acidic protein, *MMSE* mini-mental state examination, *NfL* neurofilament light chain, *p-tau181* phosphorylated-tau181, *p-tau217* phosphorylated-tau217, t-tau: total-tau.

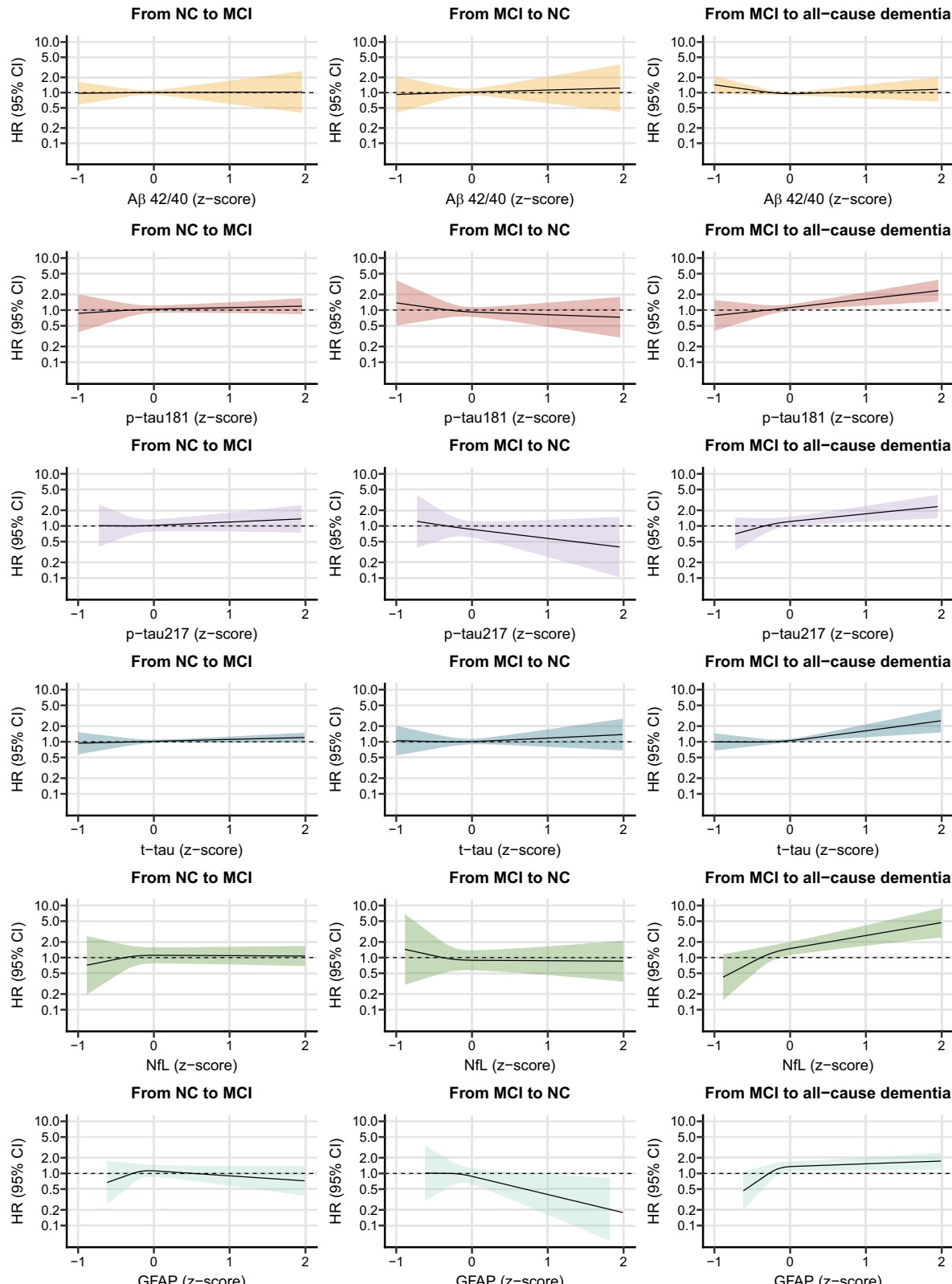

**Fig. 1 | Continuous values of blood biomarkers of Alzheimer's disease (AD) and hazard ratio (HR) of progression from normal cognition (NC) to mild cognitive impairment (MCI), reversion from MCI to NC, and progression from MCI to all-cause dementia.** Hazard Ratios (HR) with 95% Confidence Intervals (CI) are derived from multistate Markov models, using age as time scale and adjusted for sex and education. AD blood biomarkers were converted into z-scores and modeled using restricted cubic splines with 3 prespecified knots at 25th, 50th, and 75th percentiles. The median value was chosen as reference for all the z-scored biomarkers. Aβ42/40 amyloid beta 42/40, MCI mild cognitive impairment, NC normal cognition, p-tau181 phosphorylated tau 181, p-tau217 phosphorylated tau 217, t-tau total tau, NfL neurofilament light chain, GFAP glial fibrillary acidic protein. Color schemes: Aβ42/40: yellow; p-tau181: red; p-tau217: purple; t-tau: blue; NfL: green; GFAP: turquoise.

**Table 2 | Levels of blood biomarkers of Alzheimer's disease (AD) and hazard ratio (HR) of progression from normal cognition (NC) to mild cognitive impairment (MCI), reversion from MCI to NC, and progression from MCI to all-cause dementia**

| | From NC to MCI | | | From MCI to NC | | | From MCI to all-cause dementia | | |
|---|---|---|---|---|---|---|---|---|---|
| | N transitions/ participants | HR (95%CI) Basic model | HR (95%CI) Fully adjusted | N transitions/ participants | HR (95%CI) Basic model | HR (95%CI) Fully adjusted | N transitions/ participants | HR (95%CI) Basic model | HR (95%CI) Fully adjusted |
| **Aß42/40** | | | | | | | | | |
| Low vs High | 138/1119 vs 173/1029 | 0.98 (0.72, 1.34) | 0.99 (0.70, 1.41) | 129/1119 vs 164/1029 | 0.76 (0.50, 1.16) | 0.83 (0.52, 1.32) | 241/1119 vs 123/1029 | **1.30 (1.04, 1.61)** | **1.26 (1.01, 1.58)** |
| **P-tau181** | | | | | | | | | |
| High vs Low | 89/781 vs 222/1367 | 0.87 (0.59, 1.27) | 0.94 (0.58, 1.54) | 77/781 vs 216/1367 | **0.50 (0.30, 0.84)** | 0.60 (0.32, 1.13) | 219/781 vs 145/1367 | **1.36 (1.08, 1.70)** | **1.26 (1.00, 1.59)** |
| **P-tau217** | | | | | | | | | |
| High vs Low | 81/803 vs 230/1345 | 1.07 (0.73, 1.56) | 1.19 (0.75, 1.88) | 72/803 vs 221/1345 | 0.69 (0.40, 1.18) | 0.87 (0.47, 1.59) | 231/803 vs 133/1345 | **1.74 (1.38, 2.19)** | **1.64 (1.29, 2.08)** |
| **T-tau** | | | | | | | | | |
| High vs Low | 152/1090 vs 159/1058 | 1.18 (0.83, 1.66) | 0.99 (0.69, 1.42) | 133/1090 vs 160/1058 | 1.10 (0.69, 1.77) | 0.90 (0.56, 1.44) | 225/1090 vs 139/1058 | **1.42 (1.14, 1.76)** | **1.35 (1.08, 1.68)** |
| **NfL** | | | | | | | | | |
| High vs Low | 97/941 vs 214/1207 | 0.91 (0.57, 1.45) | 0.70 (0.42, 1.19) | 84/941 vs 209/1207 | **0.56 (0.31, 1.00)** | **0.41 (0.22, 0.78)** | 267/941 vs 97/1207 | **1.84 (1.43, 2.36)** | **1.64 (1.27, 2.14)** |
| **GFAP** | | | | | | | | | |
| High vs Low | 99/879 vs 212/1269 | 0.91 (0.63, 1.30) | 0.86 (0.57, 1.31) | 87/879 vs 206/1269 | **0.53 (0.33, 0.85)** | **0.50 (0.30, 0.85)** | 253/879 vs 111/1269 | **1.47 (1.17, 1.87)** | **1.35 (1.06, 1.71)** |

Hazard Ratios (HR) with 95% Confidence Intervals (CI) are derived from multistate Markov models, using age as time scale. The basic model is adjusted for sex and education; the fully adjusted model is further adjusted for chronic kidney disease, heart diseases, cerebrovascular disease, anemia and obesity. Cut-offs: 0.057 for Aß–42/40 ratio, 1.512 pg/mL for p-tau181, 0.134 pg/mL for p-tau217, 0.832 pg/mL for t-tau, 20.171 pg/mL for NfL and 142.515 pg/mL for GFAP.
*Aß42/40* amyloid beta 42/40, *MCI* mild cognitive impairment, *NC* normal cognition, *p-tau181* phosphorylated tau 181, *p-tau217* phosphorylated tau 217, *t-tau* total tau, *NfL* neurofilament light chain, *GFAP* glial fibrillary acidic protein.

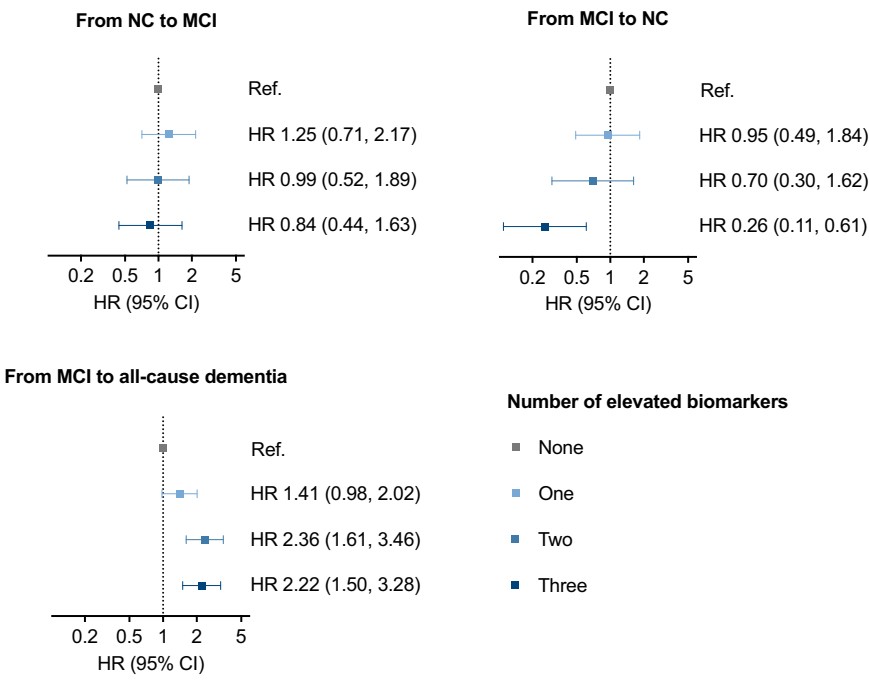

**Fig. 2 | Hazard ratio (HR) of progression from normal cognition (NC) to mild cognitive impairment (MCI), reversion from MCI to NC, and progression from MCI to all-cause dementia, depending on number of elevated blood biomarkers of Alzheimer's disease (AD), among p-tau217, NfL and GFAP.** Centre and error bars represent Hazard Ratios (HR) with 95% Confidence Intervals (CI), derived from multistate Markov models, using age as time scale and adjusted for sex, education, chronic kidney disease, heart diseases, cerebrovascular disease, anemia and obesity. Cut-offs: 0.134 pg/mL for p-tau217, 20.171 pg/mL for NfL and 142.515 pg/mL for GFAP. N transitions/participants for each group: from NC to MCI, none 148/832, one 81/449, two 50/427, three 32/440; from MCI to NC, none 143/832, one 80/449, two 47/427, three 23/440; from MCI to all-cause dementia, none 39/832, one 58/449, two 108/427, three 159/440. MCI mild cognitive impairment, NC normal cognition, p-tau217 phosphorylated tau 217, NfL neurofilament light chain, GFAP glial fibrillary acidic protein. Color schemes for number of elevated biomarkers: none: gray; one: light blue; two: medium blue; three: dark blue.

with both biomarkers elevated. For progression to AD dementia, the HRs were 2.09 (95% CI 1.37, 3.17) for high NfL only, 2.03 (95% CI 1.23, 3.36) for high p-tau217 only, and 3.07 (95% CI 2.04, 4.60) when both biomarkers were elevated.

Conversely, the hazard of MCI reversion to normal cognition decreased with the number of elevated biomarkers at baseline and was 70% lower in individuals with three compared to those with no elevated biomarkers (Fig. 2 and Supplementary Tables S10 and S11). No association was found between biomarker combinations and the progression from normal cognition to MCI (Fig. 2).

## Discussion

In this large community-based study elevated levels of several AD blood biomarkers were associated with faster cognitive deterioration. Specifically, individuals with higher p-tau181, p-tau217, t-tau, NfL, and GFAP experienced a faster progression from MCI to all-cause and AD dementia compared to those with lower biomarker levels. These associations were even stronger when multiple biomarkers were considered together. A low Aβ42/40 ratio was also associated with the progression from MCI to all-cause and AD dementia. High NfL and GFAP were associated with a less likely reversion from MCI to normal cognition. Notably, no association emerged between any of the examined biomarkers and the development of MCI among cognitively unimpaired participants.

A growing number of longitudinal studies, including one conducted by our group[9], confirmed the association between AD blood biomarkers and dementia onset in cognitively unimpaired community-dwelling older adults[5–8]. However, dementia can take years to manifest clinically in cognitively unimpaired individuals with elevated biomarkers, who may remain asymptomatic throughout their lifetime[18,19]. Our previous study suggested that while AD blood biomarkers could potentially be used to exclude impending dementia, they are not yet suitable as standalone screening tools for cognitively unimpaired individuals in the community[9].

The present study builds upon this evidence and offers a broader perspective by also examining the impact of AD blood biomarkers on the progression across different stages of cognitive decline (i.e., the development of MCI, the progression from MCI to dementia and MCI reversion) in the community. Our findings show that biomarker levels are most strongly associated with the more advanced phases of cognitive decline, while their utility for very early detection in asymptomatic populations remains limited. This result aligns with current recommendations that discourage the use of AD biomarkers in asymptomatic individuals[20,21] and with previous evidence[22] suggesting that AD blood biomarkers may be more informative at the MCI stage.

In this study, we placed specific focus on MCI, a clinically meaningful stage of cognitive decline that precedes dementia and may represent the optimal window for biomarker testing and for early therapeutic interventions, since disease-modifying therapies are expected to be most effective before the clinical manifestation of dementia[12]. Previous studies showed that AD blood biomarkers can identify individuals with MCI who are at the highest risk of progression to dementia in clinical settings[13–17,23,24]. Our study confirms these findings and extends them to the community, showing that AD blood biomarkers can help stratify the risk of progression from MCI to dementia, potentially providing valuable information to guide further research and the design of preventive and therapeutic strategies. Larger studies specifically focused on individuals with mild cognitive deficits should determine whether these biomarkers—alone or combined with other clinical data—can ultimately support the development of tools for individual-level dementia prediction in the general population.

We also observed a low tendency for reversion to normal cognition in participants with mild cognitive deficits and altered AD blood biomarkers. Previous studies highlighted the heterogeneous and dynamic nature of MCI[25], emphasizing that mild cognitive deficits are not necessarily a prodrome to dementia. In fact, a non-negligible portion of individuals with MCI experience spontaneous remission[25–27], particularly in community settings. A possible explanation is that mild cognitive deficits are not necessarily the expression of an underlying neuropathology. Instead, they can also be due to other transient causes, such as medical conditions, sleep disorders, or depression[25]. Blood biomarkers can help identify cases in which mild cognitive deficits are due to underlying neuropathological changes and are thus unlikely to revert.

In this study, we found weak associations between Aβ42/40 ratio and cognitive deterioration. Although the CSF Aβ42/40 ratio is an established AD biomarker[20], blood concentrations of Aβ do not correlate as directly with Aβ brain deposition as CSF levels. Blood Aβ levels are up to 10-fold lower than CSF levels and largely result from peripheral production[4,28]. Similar considerations limit the use of blood t-tau[29,30]. Consistent with previous findings from SNAC-K[9] and other community-based cohorts[5–8] p-tau217, NfL, and GFAP exhibited the strongest association with cognitive deterioration. Notably, p-tau217 was more strongly associated with AD dementia than all-cause dementia. This result is not unexpected, since p-tau217 is the most sensitive blood biomarker for AD pathology[31–34] and was proposed as a standalone biomarker for AD diagnosis[20]. Our findings further confirm its role, even in the community, and suggest its potential use as an initial assessment tool for AD pathology in individuals with mild cognitive deficits. P-tau217 could help determine the need for further clinical evaluation or confirmatory testing and aid in identifying candidates for disease-modifying therapies. Conversely, NfL is a nonspecific marker of neurodegeneration that broadly reflects neuronal loss[35,36]. Due to its lack of specificity, NfL may be useful to detect all-cause dementia in community settings, where most dementia cases occur in old age and are a result of mixed neuropathological processes[37–40]. Our findings on GFAP, marker of astrocytic activation, which can be a response to amyloid plaque formation in AD[41], align with evidence reporting its association with the progression from MCI to dementia[16,24].

As expected, the risk of progression to all-cause and AD dementia was even higher in individuals with multiple altered biomarkers. Among the biomarker combinations, the strongest association with the progression to dementia was observed for elevated p-tau217 with NfL. The simultaneous alteration of different blood biomarkers may reflect the presence of different neuropathological changes or indicate a more advanced disease stage. Integrating multiple biomarkers could, therefore, provide complementary information and further enhance risk stratification in individuals with mild cognitive deficits[13].

In line with our previous study in SNAC-K[9], the association between AD blood biomarkers and the progression from MCI to all-cause and AD dementia was generally slightly stronger in participants younger than 78 than in the older ones. This, along with the observation that AD blood biomarker levels increase with age, even in cognitively unimpaired individuals[42,43], suggests the potential need for age-adjusted cutoffs. In addition, the association between AD blood biomarkers and the progression to all-cause dementia appeared stronger in women than in men. However, we cannot draw definitive conclusions due to overlapping confidence intervals. Future studies should investigate deeper whether age or sex can influence the relationship between AD blood biomarkers and cognitive outcomes and explore the use of age- or sex- adjusted cut-offs.

A key strength of this study is the inclusion of over 2000 community-dwelling older adults with up to 16 years of follow-up, in whom we measured six AD blood biomarkers, including p-tau217. Additionally, the use of a neuropsychological battery, administered by trained psychologists, ensured an objective and rigorous identification of mild cognitive deficits (i.e., MCI). Finally, the combination of a long follow-up with the use of multistate Markov models enabled us to

examine the role of AD blood biomarkers in all possible transitions across the cognitive spectrum. Some limitations should also be mentioned. First, dementia diagnosis was based solely on clinical assessment and did not incorporate neuroimaging or CSF biomarkers, introducing a potential risk of AD dementia misclassification. This challenge is inherent to community-based settings, where biological confirmation is rarely available and mixed etiologies are frequent. To enhance diagnostic accuracy, we adopted a standardized three-step procedure involving two trained physicians and a neurologist with expertise in dementia diagnosis. We also reviewed clinical charts and death registers to ensure the detection of dementia in deceased participants. Second, AD biomarkers were measured in serum, where their concentration is lower compared to plasma. Nevertheless, serum biomarker levels correlate closely with plasma levels[44,45] and show comparable diagnostic performance[44,46]. Third, our results based on cut-offs derived within the SNAC-K cohort, may not fully extend to more diverse populations or alternative laboratory platforms. Future studies are needed to validate these findings in independent and more diverse cohorts. Fourth, a substantial number of participants lacked AD blood biomarkers at baseline. They were generally older, had lower educational levels, and had a higher prevalence of comorbidities compared to those included in the study. Since these individuals were at higher risk of dementia, their exclusion may have led to an underestimation of the results. Finally, the availability of biomarkers only at baseline did not allow us to assess the associations between changes in biomarker levels and progression across stages of cognitive decline.

In conclusion, in this large community-based study elevated levels of several AD blood biomarkers—particularly p-tau217, NfL, and GFAP—were associated with a faster progression from MCI to all-cause and AD dementia and with decreased MCI reversion. Notably, no association emerged between biomarker levels and the development of MCI in cognitively unimpaired participants. These findings demonstrate robust group-level associations and suggest that AD blood biomarkers may help stratify the risk of progression to dementia at the MCI stage in the community. Future studies including larger cohorts of individuals with mild cognitive deficits should explore the integration of clinical and biomarker data to improve individual-level dementia prediction.

## Methods

### Study population

We used data from the *Swedish National study on Aging and Care in Kungsholmen* (SNAC-K)[47], an ongoing longitudinal population-based study. At baseline (2001-2004), 3363 randomly selected individuals aged 60+ from the Kungsholmen district of Stockholm were enrolled (73% participation rate). Participants were followed up every six (<78 years old) or three years (≥78 years old). We selected dementia-free participants at baseline ($n = 3123$) and excluded individuals missing data on blood biomarkers ($n = 833$), obtaining a baseline population of 2290 participants. Participants with missing data were on average older, more frequently females, and had a lower educational level and a higher prevalence of chronic diseases ($p < 0.001$) than those with complete baseline data. Overall, 142 (6.2%) participants dropped out after the baseline assessment, leaving 2148 participants with available follow-up (Supplementary Fig S2). Compared to those with available follow-up data, participants who dropped out were younger (mean age difference: −7.52 years; 95% CI: −9.27, −5.76), more educated (university level: 47.2% vs. 35.4%, $p$ 0.044), and had fewer chronic diseases (mean difference: −1.00 diseases; 95% CI: −1.39, −0.60).

Written informed consent was provided by all participants, or by a proxy for individuals with cognitive impairment. The protocol for all waves of the SNAC-K study was approved by the Regional Ethical Review Board in Stockholm (Dnrs: KI 01-114, 04-929/3, Ö26-2007, 2009/595-32, 2010/447-31/2, 2013/828-31/3, 2016/730-31/1 and 2023-

02375-02), and ethical standards of the Declaration of Helsinki were followed throughout the investigation.

The results of the study are reported following the STROBE recommendations[48].

### Data collection

At each visit, participants underwent a standardized comprehensive evaluation by trained nurses, physicians, and psychologists. Clinical, laboratory, functional, and cognitive data were collected.

### Blood biomarkers of Alzheimer's disease

Peripheral venous blood samples were collected at baseline (fasting was not compulsory) and serum aliquots were stored at the Karolinska Institutet Bio Bank at −80 °C in cryogenic storage vials. Biomarkers were measured at the Affinity Proteomics Stockholm Unit (SciLifeLab). Simoa Neuro 3-plex A Kit was used for serum amyloid-β40 (Aβ40), amyloid-β42 (Aβ42) and total-tau (t-tau), Simoa pTau-181 Advantage V2 Kit for serum p-tau181, Simoa ALZpath p-Tau-217 Advantage PLUS Kit for serum p-tau217 and Simoa Neuro 2-plex B Kit for serum NfL and GFAP. For each kit, 25 μL of sample were diluted 1:4 and the assays were performed according to manufacturer instructions. The Quanterix instrument provides AEB (average enzyme per bead) values for calibrators, controls and samples. The Quanterix SR-X software automatically performs curve-fitting, extrapolation of concentrations and graphical representation using the calibrators and a four-parameter logistic (4PL) curve fit. Precision was estimated for each assay. Within run coefficient of variation (CV) was calculated on a triplicate serum pool (from SNAC-K samples), control 1 (Quanterix kit) and control 2 (Quanterix kit) included in each run (plate). The average CV for all runs is reported in Supplementary Table S14. Data below the limit of detection were replaced, using a not missing at random strategy, through single-value imputation, with a value of 0 ($n = 6$ for Aβ42, 15 for p-tau181, 5 for p-tau217 and 15 for t-tau). AD blood biomarkers were standardized into $z$-scores and modeled using restricted cubic splines with three knots (25th, 50th, and 75th percentiles). Blood biomarkers were also dichotomized (high/low) using cut-offs derived in a previous study[9]. Briefly, the study population was randomly split into a training dataset (80%) and a testing dataset (20%). Within the training dataset, we used a non-parametric bootstrapping procedure (5000 iterations) to identify the optimal cut-off for each biomarker to predict the onset of all-cause dementia within 10 years. The optimal threshold was determined by maximizing Youden's Index. These cut-offs were then applied to the testing dataset to validate their predictive performance.

### Mild cognitive impairment (MCI)

MCI was operationalized using a neuropsychological test battery[49] including five cognitive domains: executive function (Trail Making Test Part B), episodic memory (free recall), visuospatial abilities (mental rotations), language (Category and Letter Fluency), and perceptual speed (digit cancellation and pattern comparison). Test scores were standardized into $z$-scores, using the age-specific baseline mean and standard deviation (SD); if more than one cognitive test per domain was available, the $z$-scores were averaged. Individuals were classified as having MCI if they scored ≤1.5 SD below the age-specific mean in at least one cognitive domain and showed largely preserved functional independence—defined as no more than one impaired instrumental activity of daily living (IADL) and preserved basic activities of daily living (ADLs)[50]—in the absence of a dementia diagnosis. The same procedure was used to identify MCI at follow-ups, using the baseline cut-offs.

An alternative operationalization—cognitive impairment, no dementia (CIND)—was also adopted, defined as scoring ≤1.5 SD below the age-specific mean in at least one cognitive domain in the absence of a dementia diagnosis[10].

## All-cause and AD dementia diagnosis

At baseline and at each follow-up assessment, all-cause and AD dementia were diagnosed following the Diagnostic and Statistical Manual of Mental Disorders, 4th Edition, (DSM-IV) criteria[51] and the NINCDS-ADRDA criteria[52]. The diagnosis followed a three-step procedure. A first preliminary diagnosis was made by the examining physician, who conducted the in-person clinical assessment. A second physician, blinded to the initial assessment, independently reviewed the clinical documentation, and made a second preliminary diagnosis. In the case of disagreement between the two, a senior neurologist with expertise in cognitive disorders—external to the data collection process—reviewed the case and issued a final diagnosis. If participants died without a dementia diagnosis, further information was obtained by reviewing clinical charts and the Swedish National Cause of Death Register. This approach minimizes the risk of underestimating dementia cases due to death occurring before a scheduled follow-up assessment.

## Vital status

Information on the participants vital status was obtained from the Swedish Cause of Death Register.

## Other covariates

Education was categorized into elementary, high school, and university or higher. The Mini-Mental State Examination was used as a measure of global cognition. Information on chronic diseases was collected by physicians through clinical evaluation, medical records, self-reports, laboratory tests, and medication use. Diseases were coded following the International Classification of Diseases 10th revision (ICD-10)[53].

## Statistical analysis

Parametric multistate Markov models[54] were used to estimate the hazard ratios (HR) for transition between cognitive stages, according to the levels of AD blood biomarkers and the combination of elevated biomarkers at baseline. These models provide the opportunity to explore the association between AD biomarkers and all possible transitions between cognitive stages simultaneously. We used a four-state model accounting for interval-censoring. We included three cognitive stages (normal cognition, MCI, and dementia) and death as an absorbing state, meaning that individuals who reach the death stage cannot move back to any of the previous stages. The direct transition from normal cognition to dementia was not allowed; individuals who progressed directly from normal cognition to dementia were assumed to have passed through a MCI phase, although this was not observed due to intermittent observation. The only reversible transition was the one between MCI and normal cognition. The diagram of the adopted multistate Markov model is depicted in Supplementary Fig S3. For participants who were alive and dementia-free but lacked information on their MCI status, the model accounted for this uncertainty by assuming that their exact state was unknown but could be limited to either normal cognition or MCI. When dementia was diagnosed at death, we considered that dementia onset occurred between the time of last assessment and the time of death.

The baseline transition hazards of the model were assumed to follow a Gompertz distribution[55], with age used as time scale. The analyses were adjusted for sex and education; additional adjustments included chronic kidney disease, heart disease, cerebrovascular diseases, anemia, and obesity, as we previously found that AD blood biomarker levels vary with these comorbidities[42].

## Subgroup and sensitivity analyses

Models with age-dependent (i.e., ≤78 and >78 years old) HRs for the biomarkers were also fitted. The analyses were also stratified by sex. To assess whether including individuals with MCI at baseline could have influenced the results, since the duration of MCI status was unknown, the analyses were repeated excluding those with MCI at baseline ($n = 381$). To account for potential bias due to attrition, sensitivity analyses using inverse probability weighting (IPW) were conducted, with weights derived from logistic regression models including age, sex, education, number of chronic diseases, and biomarker levels. We repeated all main analyses adopting the operationalization of CIND.

Statistical analyses were conducted using R version 4.3.1 (The R Foundation for Statistical Computing). To fit multistate models we used the msm[56] package.

## Reporting summary

Further information on research design is available in the Nature Portfolio Reporting Summary linked to this article.

## Data availability

SNAC-K data are sensitive data; thus, they cannot be shared publicly, but raw and analyzed de-identified data can be requested by qualified researchers at https://www.snac-k.se/. The request will be reviewed to ensure confidentiality and intellectual-property obligations. A data-sharing agreement must be signed prior to data release. Source data are provided with this paper.

## Code availability

Analysis codes for this study are available at https://github.com/ARCbiostat/biomdemstages/tree/main.

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

## Acknowledgements

The authors acknowledge the SNAC-K participants and the SNAC-K group for their collaboration in data collection and management, and the Affinity Proteomics-Stockholm unit at SciLifeLab for supporting with biomarkers quantification in serum. Data collection of the Swedish National study on Aging and Care (SNAC-K) was supported by the Swedish Research Council (current grant: 2021-00178); the Swedish Ministry of Health and Social Affairs; the participating County Councils and Municipalities. This work was further supported by Stiftelsen Sigurd och Elsa Goljes minne (M.V., Project No.: LA2024-0126), Demensfonden (M.V., 2024) Hjärnfonden (postdoc stipend to G.G.; 2021-0025) and Gamla Tjänarinnor foundation (G.G. 2021-01235); D.L.V. was supported by the Swedish Research Council (project number 2021-03324) and the Karolinska Institutet Strategic Research Area in Epidemiology and Biostatistics (SFOepi) in 2021 and 2023. EJL was supported by the Swedish Research Council (project numbers 2020-01030; 2024-00721). The funders had no role in study design, data collection, data analyses, data interpretation or writing of the report.

## Author contributions

M.V., D.L.V., and G.G. contributed to the conception and design of the study. C.F., M.D., and S.A. conducted the biomarkers' analyses. M.V. and C.G. conducted the statistical analyses. M.V., D.L.V., G.G., C.G., D.R., B.W., M.C., E.J.L. and L.F. contributed to interpretation of the results. M.V. drafted the first version of the manuscript. All authors critically revised the manuscript for important intellectual content. All authors made a significant contribution to the research and the development of the manuscript and approved the final version for publication.

## Funding

## Competing interests

The authors declare no competing interests.
