## [Transparent Peer Review file · Nature Communications]

Blood biomarkers of Alzheimer's disease and progression across different stages of cognitive decline in the community

Corresponding Author: Dr Martina Valletta

Version 0:

Reviewer comments:

Reviewer #1

(Remarks to the Author)

1, This study used SNAC-K to examine the association between baseline levels of AD blood biomarkers and transitions across cognitive stages: normal cognition; cognitive impairment no dementia (CIND), and all-cause dementia. However, the same team has published a paper 'Grande, G. et al. Blood-based biomarkers of Alzheimer's disease and incident dementia in the community. Nat. Med. (2025) doi:10.1038/s41591-025-03605-x', which is similar to this paper. Also many previous studies have already verified the associations between AD blood biomarkers and dementia/MCI (as the outcome), even with larger sample size. Therefore this paper lacked novelty.

2, CIND is just an unstable stage between normal cognition and dementia, and there's no golden standard pathological diagnosis criteria, e.g. for MCI due to AD, the pathological biomarker is Ab PET. Therefore there's no proper rationale to analyze NC-CIND and CIND-NC.

3, Figure 4: The authors used "number of elevated biomarkers" and "combination of 3 biomarkers" for the subgroup analysis, however, it lacked clinical/pathological evidence for the grouping definitions. E.g. low ptau-217 and low NfL as the reference, but there are groups with 'low ptau-217 and high NfL', 'high ptau-217 and low NfL', which were ignored in the analysis.

Reviewer #2

(Remarks to the Author)

Valletta et al. investigated AD-related blood-based biomarkers in a population-based cohort of 2148 dementia-free individuals, assessing whether these biomarkers were associated with changes in cognitive status. They found that elevated plasma levels of p-tau181, p-tau217, t-tau, NfL, and GFAP were linked to progression from cognitive impairment no dementia (CIND) to all-cause and AD dementia. Among these, NfL and p-tau217 showed the strongest associations. In contrast, biomarker levels were not associated with the development of CIND among cognitively unimpaired participants.

The main strengths of the study are the large sample size, long follow-up period, population-based design, and the inclusion of several blood-based biomarkers. The main limitation is the absence of reference standard biomarkers such as CSF or PET imaging.

Major points:

1. The authors should clearly explain the novel contribution of this study compared to their recently published paper (Grande et al., Nat Med 2025).
2. A clearer definition of "cognitive impairment no dementia" is needed. Is this term equivalent to mild cognitive impairment (MCI)? If not, how does it differ?
3. How was the "cognitively unimpaired" group defined? What neurological and cognitive assessments were conducted to rule out any underlying neurological disease or cognitive impairment? Were any biomarkers used to detect preclinical AD?
4. Was AD dementia defined solely based on clinical criteria? Were any CSF or PET biomarkers available to support the

diagnoses?

5. The study reports a dropout rate of 6.2% (142 participants). How were these individuals handled in the analysis? Were any sensitivity analyses performed?

6. While the study clearly shows biomarker associations with cognitive progression at the group level, the authors should address whether these findings are translatable to the individual level. This is critical to support the conclusion that "individuals with mild cognitive deficits" are an appropriate target for community-based biomarker testing.

Reviewer #3

(Remarks to the Author)

Thank you for the opportunity to review the above manuscript and I commend the authors for their detailed exploration of the relationship between BBM of AD and progression in dementia free adults.

I feel that this would be a much more impactful manuscript if authors could change from analysing z-scores of biomarkers to predict progression in CIND to biomarker "positivity" (particularly for p-tau217) in MCI - this in my mind would have a significant impact on the applicability of these findings to a broader audience and in particular is an incredibly useful clinical question. This would require: (i) cleaning up definition of CIND/MCI to reflect individuals who are functionally intact (using 1.5SD below the mean as cut off is acceptable in my view, this small step would 'complete' the definition) and (ii) defining cut off values for p-tau217, p-tau181 and ratios at least to give us HR for conversion in MCI with positive blood biomarkers.

Notable points:

- The study has over two thousand participants, from the SNAC-K cohort. As far as I'm aware there is limited data in these types of cohorts using BBMs, so this is commendable.
- Study is commendable for 16 year follow up. Many studies have only examined short term
- Also commendable to look at a panel of BBMs, not just leading BBM.
- Study setting and recruitment well described
- The description of biomarker analysis using Simoa is excellent; I cannot comment specifically on the spline modelling as I haven't carried out this analysis myself, but the z-scoring makes sense and description of the modelling is accurate and appropriate.
- I welcome the sex specific results, mirroring recent data that demonstrates higher levels of AD biomarkers in females.

Points for reflection/discussion

- CIND is not necessarily a well-defined term ? I would suggest using MCI as this is better understood , I would refer authors to the Manchester consensus on MCI, published in recent years in age and ageing by ross dunne and colleagues.
 - From the methods section, it appears that CIND is 1.5 SD below norms on neuropsych function - this is the exact definition of MCI. <https://pubmed.ncbi.nlm.nih.gov/33197937/>
 - Does SNAC-K have an ADL questionnaire - this should be added to determine if those with MCI/CIND are functionally intact (may lead to excluding a tiny minority of participants, but clinically this would make a lot more sense).
 - Were all individuals examined for dementia at all study visits ? This should be made clear.
 - Did authors look at p-tau217/amyloid ratios - given recent FDA approval of lumipulse for this ratio
- Individuals are characterised with z-score of biomarker. This is definitely an acceptable way to perform this analysis. However, what would really increase the clinical applicability of the findings is to get a HR for biomarker positivity. This particularly applies to p-tau217. If the authors have established cut-offs in the same lab on the same machine (for instance with participants with paired blood/CSF available - or from an existing biobank that could be run), define positivity cut-offs and use these to predict conversion

Version 1:

Reviewer comments:

Reviewer #2

(Remarks to the Author)

The authors successfully addressed all my comments. I believe that the manuscript can be accepted in its current form.

Reviewer #3

(Remarks to the Author)

I am satisfied that all my concerns have been addressed in the revision - the authors should be commended on their work

Reviewer #4

(Remarks to the Author)

Reviewer #1

- 1) Novelty vs. prior SNAC-K/Nat Med (Grande et al., Nat Med 2025) paper: In the rebuttal and revision the authors (i) clarify that the prior paper modeled incident dementia only, whereas this study uses multistate Markov models to follow all transitions (NC→MCI, MCI→NC, MCI→dementia) over up to 16 years, and (ii) explicitly re-frame the contribution as showing where blood biomarkers are most informative (i.e., at the MCI stage) in a community cohort. They expanded Intro/Discussion accordingly. This is a reasonable and, in my view, sufficient differentiation; while the novelty is incremental, it is real.
- 2) Use of CIND vs MCI: The authors re-ran the set of analysis using a standard MCI operationalization that incorporates ADL/IADL (≤ 1 impaired IADL; preserved BADLs) and show the main conclusions are unchanged. The Methods now detail this change. It was adequately addressed.
- 3) Clarity of the biomarker-combination analysis: The authors explain they did model four groups for each pair (low/low; high NfL only; high p-tau217 only; high/high), removed the confusing panel, and provided the full estimates in Supplementary Tables. It was adequately addressed.
- 4) Clinical (not biomarker) dementia ascertainment: The revision now describes the three-step clinical adjudication and openly acknowledges the lack of CSF/PET as a limitation. This is transparent and appropriate for a community cohort.

Reviewer #3

- 1) Analyze positivity in MCI (esp. p-tau217) rather than z-scores in CIND: They switched from CIND to MCI and added analyses using dichotomized biomarker cutoffs (including p-tau217) previously derived within SNAC-K (Youden's index on a training set with hold-out testing), reporting HRs for progression. This directly meets the request. (E.g., high p-tau217 and high NfL each predict faster MCI→AD dementia; effects are strongest when both are elevated.)
- 2) Clarify MCI definition and functional intactness: They now require ≤ 1.5 SD impairment in ≥ 1 domain and preserved function (ADL/IADL) per Manchester consensus; applied at baseline and follow-ups. It was adequately addressed.
- 3) Consider the p-tau217/A β 42 ratio: In response, they computed p-tau217/A β 42 and repeated the transition analyses (tables/figures shown in rebuttal). This was responsive.
- 4) Clinical diagnosis details & external validity: They added detail on the three-step adjudication, noted the lack of CSF/PET, and flagged ongoing registry linkage, suggesting high concordance of dementia diagnoses—appropriately framed as supportive but preliminary.
- 5) Avoid implying individual-level prediction: They explicitly removed individual-prediction language, restricting claims to group-level risk stratification. This was responsive.

My own thoughts

- 1) This is a strong community-based analysis with long follow-up, but the sample's diversity and external validity are limited: SNAC-K is a single urban Swedish cohort with high educational attainment (35.4% university) and a majority of women (61.5%); race/ethnicity is not reported in the baseline table, and the study setting is a relatively homogeneous, high-resource health system. As a result, performance and cut-points for blood biomarkers derived internally via an 80/20 split on this same cohort and a specific Simoa/ALZpath platform may not transfer to more diverse populations by race/ethnicity, socioeconomic status, rurality, or to different laboratory platforms without recalibration.
- 2) Methodologically, the multistate Markov framework is a strength, but several assumptions and design features temper robustness: (i) transitions are observed at visits, with the model assuming no direct normal to dementia progression (unobserved MCI is imputed), which can misclassify short prodromal intervals; (ii) biomarkers were measured once at baseline, so dynamic changes or regression dilution were not captured; and (iii) dementia diagnoses rely on clinical criteria without CSF/PET confirmation in a setting with common mixed pathologies, inviting outcome misclassification despite a careful three-step adjudication and register linkage.
- 3) Selection also matters: 833 otherwise eligible participants lacked baseline biomarkers and were older, less educated, and sicker, and 6.2% dropped out after baseline; both patterns could bias associations (likely toward underestimation per authors' discussion), although IPW analyses were performed and mostly aligned with the main results.
- 4) Finally, while findings for MCI→dementia were consistent (strongest for p-tau217, NfL, GFAP), some effects attenuate in fully adjusted and sensitivity models (e.g., p-tau181 for MCI reversion, and certain associations when baseline MCI is excluded), underscoring that individual-level prognostication is not yet supported and that broader, multi-site validation with standardized assays and pre-specified (potentially age/sex-specific) thresholds is still needed before clinical deployment in heterogeneous populations.

Point by point response to comments from the Reviewers

Reviewer #1:

A note to the attention of the Reviewer:

In response to the comments raised by Reviewers #2 and #3, we have revised the operational definition of cognitive impairment used in our study. Specifically, we replaced the construct “cognitive impairment, no dementia (CIND)” —which was based solely on lower cognitive performance, exceeding the expected level for the individual’s age—with “mild cognitive impairment (MCI)”, which additionally incorporates information on the individual’s functional ability. We have applied this new operational definition throughout the revised manuscript and rerun all the analyses accordingly.

Reviewer’s comment 1: This study used SNAC-K to examine the association between baseline levels of AD blood biomarkers and transitions across cognitive stages: normal cognition; cognitive impairment no dementia (CIND), and all-cause dementia. However, the same team has published a paper 'Grande, G. et al. Blood-based biomarkers of Alzheimer’s disease and incident dementia in the community. Nat. Med. (2025) doi:10.1038/s41591-025-03605-x', which is similar to this paper. Also many previous studies have already verified the associations between AD blood biomarkers and dementia/MCI (as the outcome), even with larger sample size. Therefore this paper lacked novelty.

Author’s reply: We appreciate this observation, which gives us the opportunity to better clarify the novelty and scope of our study in relation to previous work, including our own.

In our previously published work (Grande et al., Nat Med 2025), we examined the association between Alzheimer’s disease (AD) blood biomarkers and incident dementia in dementia-free older adults from the population-based study SNAC-K. That work was the first to show the predictive value of blood p-tau217, alongside other blood biomarkers like NfL and GFAP, offering key insights into their clinical utility for dementia prediction in a population setting. However, its focus was limited to the onset of overt dementia, without accounting for intermediate stages of cognitive decline.

In contrast, the current study specifically addresses this gap by:

1. **Adopting a multistate approach to model transitions across the entire cognitive spectrum**, including progression to MCI, reversion to normal cognition, and progression from MCI to dementia. This modeling strategy, coupled with a longitudinal follow-up of up to 16 years, enables a dynamic understanding of how biomarker levels relate to distinct cognitive trajectories. To our knowledge, this is the first application of this framework in a large community cohort. Our findings show that biomarker levels are most strongly associated with transitions involving advanced stages of decline — particularly progression from MCI to dementia — suggesting that their utility in the community may be greatest in the presence of mild cognitive deficits.
2. **Placing a specific focus on MCI in the general population**, which was not addressed in our previous publication. Prior studies exploring the association between AD biomarkers and MCI progression *were primarily conducted in clinical specialistic cohorts*. In contrast, evidence from large, community-based samples is lacking. Using SNAC-K we provide unique population-based evidence on the potential value of AD blood biomarkers in the general population. Our findings suggest that AD blood biomarkers can help risk stratification at the MCI stage even in the community.

Stimulated by the Reviewer’s comment, we have expanded the Introduction and the Discussion sections of the revised manuscript to better highlight the elements of novelty of our study and how it builds upon – and differs from – previous work, including our own. We have also enriched the References regarding previous studies on the topic.

In the Introduction:

“In line with this evidence, we recently demonstrated that phosphorylated tau (p-tau) isoforms, neurofilament light chain (NfL), and glial fibrillary acidic protein (GFAP) can accurately predict future all-cause and AD dementia in a community-based cohort⁹. However, evidence regarding their association with progression across intermediate stages of cognitive decline remains limited.

[...]

While a few clinical studies have shown that individuals with mild cognitive impairment (MCI) and elevated levels of AD blood biomarkers are at high risk of progression to dementia¹³⁻¹⁷, data from population-based studies are still lacking. Moreover, how AD blood biomarkers influence the transitions across different stages of cognitive decline, including the development of MCI, the reversion to normal cognition and the progression to dementia, remains poorly understood.”

In the Discussion:

“The present study builds upon this evidence and offers a broader perspective by also examining the impact of AD blood biomarkers on the development of MCI, on the progression from MCI to dementia and on MCI reversion. To the best of our knowledge, this is the first study that explored the association between AD blood biomarkers and the progression across different stages of cognitive decline in the community.

[...]

In this study, we placed specific focus on MCI, a clinically meaningful stage of cognitive decline that precedes dementia and may represent the optimal window for biomarker testing and for early therapeutic interventions, since disease-modifying therapies are expected to be most effective before the clinical manifestation of dementia¹².”

Reviewer’s comment 2: CIND is just an unstable stage between normal cognition and dementia, and there's no golden standard pathological diagnosis criteria, e.g. for MCI due to AD, the pathological biomarker is Ab PET. Therefore there's no proper rationale to analyze NC-CIND and CIND-NC.

Author’s reply: As the Reviewer pointed out, we decided to adopt a clinically based definition of CIND/MCI in our analyses. Prompted by the comments of Reviewers#2 and #3 we have revised the operationalization of cognitive impairment in our study and adopted MCI, which includes the assessment of functional ability and independence in daily living, instead of CIND. Importantly, the results remained consistent with our previous analyses.

While we understand the Reviewer’s concern regarding the lack of gold-standard biomarker confirmation, such as amyloid PET, we believe that clinically defined cognitive outcomes, including MCI and dementia, remain meaningful and clinically relevant. This is in line with previous studies, conducted both in population-based and specialized settings (e.g., Dunne et al., *Age Ageing*, 2021; Lu et al. *JAMA* 2024; Grande et al. *Nature Medicine* 2025).

Our choice is further supported by the following considerations:

- **Heterogeneity and mixed pathologies in MCI:** Like dementia, MCI in older adults frequently reflects mixed underlying pathologies, rather than pure AD. Several autopsy studies have demonstrated that even amnesic MCI is pathologically heterogeneous, often involving co-existing vascular, Lewy body, or TDP-43 pathology (e.g., Schneider JA et al., *Ann Neurol* 2009). In this context, a purely biomarker-confirmed definition of AD-MCI would not capture the actual complexity of cognitive decline in aging populations. Our focus on clinical MCI, instead, mirrors the heterogeneity and complexity of cognitive impairment in the community.
- **Prioritizing early detection beyond AD pathology:** Equally crucial to detecting "AD pathology" is the timely identification of cognitive impairment that, even in its mild stages, can affect the daily lives of individuals and their families. Identifying mild cognitive changes — regardless of their exact pathological substrate — is essential for timely support, treatment, and preventive strategies. The availability of targeted therapies for AD pathology is just one of many reasons to pursue the timely and accurate detection of MCI.

- **Biomarker specificity:** While phosphorylated tau isoforms such as p-tau217 and p-tau181 are considered specific for AD pathology, other markers used in our study (e.g., NfL, GFAP) reflect broader neurodegenerative or astroglial processes and are not AD-specific. This makes them particularly informative for capturing mild cognitive changes across different dementia subtypes, reinforcing the appropriateness of using clinically defined cognitive transitions as outcomes.
- **MCI as an unstable construct:** As the Reviewer noted, CIND/MCI is a heterogeneous and often unstable clinical construct, with a substantial proportion of individuals remaining cognitively stable or even reverting to normal cognition over time (Canevelli et al., JAMDA 2016; Salemme et al., *Alzheimers Dement* 2025). This unpredictability is precisely what makes the identification of scalable and accessible biomarkers for the progression to dementia so important, especially where PET or CSF testing is not feasible.

These considerations were incorporated into the Discussion section of the revised manuscript. For example:

“We also observed a low tendency for reversion to normal cognition in participants with mild cognitive deficits and altered AD blood biomarkers. Previous studies highlighted the heterogeneous and dynamic nature of MCI²⁵, emphasizing that mild cognitive deficits are not necessarily a prodrome to dementia. In fact, a non-negligible portion of individuals with MCI experience spontaneous remission^{25–27}, particularly in community settings. A possible explanation is that mild cognitive deficits are not necessarily the expression of an underlying neuropathology. Instead, they can also be due to other transient causes, such as medical conditions, sleep disorders, or depression²⁵. Blood biomarkers can help identify cases in which mild cognitive deficits are due to underlying neuropathological changes and are thus unlikely to revert.”

Reviewer’s comment 3: Figure 4: The authors used "number of elevated biomarkers" and "combination of 3 biomarkers" for the subgroup analysis, however, it lacked clinical/pathological evidence for the grouping definitions. E.g. low ptau-217 and low NfL as the reference, but there are groups with 'low ptau-217 and high NfL', 'high ptau-217 and low NfL', which were ignored in the analysis.

Author’s reply: We thank the Reviewer for pointing out that Figure 2B may have appeared unclear. In the analyses examining the combined effect of biomarker pairs (e.g., p-tau217 and NfL), we categorized participants into four groups based on the levels of the two biomarkers (e.g., low p-tau217 and low NfL; high NfL only, high p-tau217 only, and both biomarkers elevated). We then estimated hazard ratios (HRs) for each cognitive transition using the group with low levels of both biomarkers as the reference.

From these analyses, we found that **compared to individuals with low levels of both biomarkers:**

- Those with **high NfL only** had an increased hazard of progression from MCI to all-cause dementia (HR 1.59, 95% CI 1.13, 2.22) and AD dementia (HR 2.09, 95% CI 1.37, 3.17).
- Similarly, individuals with **high p-tau217 only** showed an increased hazard of progression from MCI to all-cause dementia (HR 1.71, 95% CI 1.11, 2.63) and AD dementia (HR 2.03, 95% CI 1.23, 3.36).
- **The risk was highest for those with both biomarkers elevated** (HR 2.29, 95% CI 1.62, 3.24 and HR 3.07, 95% CI 2.04, 4.60, respectively).

Similar results were obtained for the other combinations of biomarkers.

Because the figure did not convey this clearly, we removed it, clarified the text, and provided full results in **Supplementary Tables S12 and S13**.

Reviewer #2:

Valletta et al. investigated AD-related blood-based biomarkers in a population-based cohort of 2148 dementia-free individuals, assessing whether these biomarkers were associated with changes in cognitive status. They found that elevated plasma levels of p-tau181, p-tau217, t-tau, NfL, and GFAP were linked to progression from cognitive impairment no dementia (CIND) to all-cause and AD dementia. Among these, NfL and p-tau217 showed the strongest associations. In contrast, biomarker levels were not associated with the development of CIND among cognitively unimpaired participants.

The main strengths of the study are the large sample size, long follow-up period, population-based design, and the inclusion of several blood-based biomarkers. The main limitation is the absence of reference standard biomarkers such as CSF or PET imaging.

We thank the Reviewer for the comments, which have significantly improved our paper, and for highlighting the main strengths of our study. As a note, in response to the comments raised by the Reviewers, we have revised the operational definition of cognitive impairment used in our study. Specifically, we replaced the construct “cognitive impairment, no dementia (CIND)” — which was based solely on lower cognitive performance exceeding the expected level for the individual’s age — with “mild cognitive impairment (MCI)”, which additionally incorporates information on the individual’s functional ability. We have applied this new operational definition throughout the revised manuscript and rerun all the analyses accordingly. A point-by-point response to the Reviewer’s comments follows below.

Major points:

Reviewer’s comment 1: The authors should clearly explain the novel contribution of this study compared to their recently published paper (Grande et al., Nat Med 2025).

Author’s reply: We appreciate this observation, which gives us the opportunity to better clarify the novelty and scope of our study in relation to previous work, including our own.

In our previously published work (Grande et al., Nat Med 2025), we examined the association between Alzheimer’s disease (AD) blood biomarkers and incident dementia in dementia-free older adults from the population-based study SNAC-K. That work was the first to show the predictive value of blood p-tau217, alongside other blood biomarkers like NfL and GFAP, offering key insights into their clinical utility for dementia prediction in a population setting. However, its focus was limited to the onset of overt dementia, without accounting for intermediate stages of cognitive decline.

In contrast, the current study specifically addresses this gap by:

1. **Adopting a multistate approach to model transitions across the entire cognitive spectrum**, including progression to MCI, reversion to normal cognition, and progression from MCI to dementia. This modeling strategy, coupled with a longitudinal follow-up of up to 16 years, enables a dynamic understanding of how biomarker levels relate to distinct cognitive trajectories. To our knowledge, this is the first application of this framework in a large community cohort. Our findings show that biomarker levels are most strongly associated with transitions involving advanced stages of decline—particularly progression from MCI to dementia— suggesting that their utility in the community may be greatest in the presence of mild cognitive deficits.
2. **Placing a specific focus on MCI in the general population**, which was not addressed in our previous publication. Prior studies exploring the association between AD biomarkers and MCI progression *were primarily conducted in clinical specialist cohorts*. In contrast, evidence from large, community-based samples is lacking. Using SNAC-K we provide unique population-based evidence on the potential value of AD blood biomarkers in the general population. Our findings suggest that AD blood biomarkers can help risk stratification at the MCI stage even in the community.

Stimulated by the Reviewer’s comment, we have expanded the Introduction and the Discussion sections of the revised manuscript to better highlight the elements of novelty of our study and how it builds upon – and differs from – previous work, including our own. We have also enriched the References regarding previous studies on the topic.

In the Introduction:

“In line with this evidence, we recently demonstrated that phosphorylated tau (p-tau) isoforms, neurofilament light chain (NfL), and glial fibrillary acidic protein (GFAP) can accurately predict future all-cause and AD dementia in a community-based cohort⁹. However, evidence regarding their association with progression across intermediate stages of cognitive decline remains limited.

[...]

While a few clinical studies have shown that individuals with mild cognitive impairment (MCI) and elevated levels of AD blood biomarkers are at high risk of progression to dementia^{13–17}, data from population-based studies are still lacking. Moreover, how AD blood biomarkers influence the transitions across different stages of cognitive decline, including the development of MCI, the reversion to normal cognition and the progression to dementia, remains poorly understood.”

In the Discussion:

“The present study builds upon this evidence and offers a broader perspective by also examining the impact of AD blood biomarkers on the development of MCI, on the progression from MCI to dementia and on MCI reversion. To the best of our knowledge, this is the first study that explored the association between AD blood biomarkers and the progression across different stages of cognitive decline in the community.

[...]

In this study, we placed specific focus on MCI, a clinically meaningful stage of cognitive decline that precedes dementia and may represent the optimal window for biomarker testing and for early therapeutic interventions, since disease-modifying therapies are expected to be most effective before the clinical manifestation of dementia¹².”

Reviewer’s comment 2: A clearer definition of “cognitive impairment no dementia” is needed. Is this term equivalent to mild cognitive impairment (MCI)? If not, how does it differ?

Author’s reply: “CIND” refers to the presence of an objective cognitive impairment that is clinically detectable through neuropsychological assessment but does not meet the diagnostic criteria for dementia (Graham JE et al., Lancet 1997). While conceptually similar to MCI as an intermediate stage between normal cognition and dementia, CIND does not require preserved functional independence. This means that individuals classified as CIND may show mild limitations in daily functioning, unlike MCI which requires that activities of daily living remain essentially intact (Dunne et al., Age Ageing, 2021, Winblad B et al., J Intern Med, 2004).

	CIND	MCI
Presence of objective* cognitive impairment that exceeds age specific means	✓	✓
Absence of dementia	✓	✓
Essentially preserved activities of daily living		✓

*Objective cognitive impairment indicates an impairment detected through a cognitive battery exploring multiple cognitive domains.

CIND is a construct often used in epidemiological studies (Graham JE et al., Lancet 1997; Grande G et al., Alzheimers Dement, 2020; Palmer K et al., Am J Psychiatry 2002; Plassman BL et al., Ann Neurol 2011; Kuo C et al., Neurobiol Aging 2023), but we acknowledge that it may be less familiar to the broader clinical and research community compared to MCI. To improve clarity and consistency with commonly used clinical terminology—and in response to a similar suggestion of Reviewer #3—we have now revised the manuscript to adopt the operationalization of MCI throughout and rerun the analyses using the MCI operationalization.

The definition of MCI has been described in the Methods section of the revised manuscript: *“MCI was operationalized using a neuropsychological test battery⁴⁹ including five cognitive domains: executive function (Trail Making Test Part B), episodic memory (free recall), visuospatial abilities (mental rotations), language (Category and Letter Fluency), and perceptual speed (digit cancellation and pattern comparison). Test scores were standardized into z-scores, using the age-specific baseline mean and standard deviation (SD); if more than one cognitive test per domain was available, the z-scores were averaged. Individuals were classified as having MCI if they scored ≤ 1.5 SD below the age-specific mean in at least one cognitive domain and showed largely preserved functional independence – defined as no more than one impaired instrumental activity of daily living (IADL) and preserved basic activities of daily living (ADLs)⁵⁰ – in the absence of a dementia diagnosis. The same procedure was used to identify MCI at follow-ups, using the baseline cut-offs.”*

Of note, the main findings of our study remain largely unchanged when using this MCI-based operationalization, and the overall interpretation of the results is consistent with our original analyses. Specifically:

- Individuals with higher p-tau181, p-tau217, t-tau, NfL, and GFAP experienced a faster progression from MCI to all-cause and AD dementia compared to those with lower biomarker levels.
- A low A β 42/40 ratio was also associated with the progression from MCI to all-cause and AD dementia.
- High NfL and GFAP were associated with a less likely reversion from MCI to normal cognition.
- No association emerged between any of the examined biomarkers and the development of MCI among cognitively unimpaired participants.
- The associations with MCI progression/reversion were even stronger when multiple biomarkers were considered together.

(See **Table** in the following page).

Table 2. Levels of blood biomarkers of Alzheimer’s disease (AD) and hazard ratio (HR) of progression from normal cognition (NC) to mild cognitive impairment (MCI), reversion from MCI to NC, and progression from MCI to all-cause dementia.

	From NC to MCI			From MCI to NC			From MCI to all-cause dementia		
	N	HR	HR	N	HR	HR	N	HR	HR
	transitions/ participants	(95%CI) Basic model	(95%CI) Full adj.	transitions/ participants	(95%CI) Basic model	(95%CI) Full adj.	transitions/ participants	(95%CI) Basic model	(95%CI) Full adj.
Aβ42/40									
Low vs High	138/1119 vs 173/1029	0.98 (0.72, 1.34)	0.99 (0.70, 1.41)	129/1119 vs 164/1029	0.76 (0.50, 1.16)	0.83 (0.52, 1.32)	241/1119 vs 123/1029	1.30 (1.04, 1.61)	1.26 (1.01, 1.58)
P-tau181									
High vs Low	89/781 vs 222/1367	0.87 (0.59, 1.27)	0.94 (0.58, 1.54)	77/781 vs 216/1367	0.50 (0.30, 0.84)	0.60 (0.32, 1.13)	219/781 vs 145/1367	1.36 (1.08, 1.70)	1.26 (1.00, 1.59)
P-tau217									
High vs Low	81/803 vs 230/1345	1.07 (0.73, 1.56)	1.19 (0.75, 1.88)	72/803 vs 221/1345	0.69 (0.40, 1.18)	0.87 (0.47, 1.59)	231/803 vs 133/1345	1.74 (1.38, 2.19)	1.64 (1.29, 2.08)
T-tau									
High vs Low	152/1090 vs 159/1058	1.18 (0.83, 1.66)	0.99 (0.69, 1.42)	133/1090 vs 160/1058	1.10 (0.69, 1.77)	0.90 (0.56, 1.44)	225/1090 vs 139/1058	1.42 (1.14, 1.76)	1.35 (1.08, 1.68)
NfL									
High vs Low	97/941 vs 214/1207	0.91 (0.57, 1.45)	0.70 (0.42, 1.19)	84/941 vs 209/1207	0.56 (0.31, 1.00)	0.41 (0.22, 0.78)	267/941 vs 97/1207	1.84 (1.43, 2.36)	1.64 (1.27, 2.14)
GFAP									
High vs Low	99/879 vs 212/1269	0.91 (0.63, 1.30)	0.86 (0.57, 1.31)	87/879 vs 206/1269	0.53 (0.33, 0.85)	0.50 (0.30, 0.85)	253/879 vs 111/1269	1.47 (1.17, 1.87)	1.35 (1.06, 1.71)

Hazard Ratios (HR) with 95% Confidence Intervals (CI) are derived from multistate Markov models, using age as time scale. The basic model is adjusted for sex and education; the fully adjusted model is further adjusted for chronic kidney disease, heart diseases, cerebrovascular disease, anemia and obesity. Cut-offs: 0.057 for Aβ-42/40 ratio, 1.512 pg/mL for p-tau181, 0.134 pg/mL for p-tau217, 0.832 pg/mL for t-tau, 20.171 pg/mL for NfL and 142.515 pg/mL for GFAP. Abbreviations: Aβ42/40: amyloid beta 42/40; MCI: mild cognitive impairment; NC: normal cognition; p-tau181: phosphorylated tau 181; p-tau217: phosphorylated tau 217; t-tau: total tau; NfL: neurofilament light chain; GFAP: glial fibrillary acidic protein.

Reviewer’s comment 3: How was the “cognitively unimpaired” group defined? What neurological and cognitive assessments were conducted to rule out any underlying neurological disease or cognitive impairment? Were any biomarkers used to detect preclinical AD?

Author’s reply: We thank the Reviewer for this comment, which gives us the opportunity to clarify the methodology used to define cognitively unimpaired participants.

As mentioned above, at baseline and at each follow-up assessment, all participants underwent a neuropsychological evaluation conducted by trained psychologists. This battery assessed five cognitive domains: executive function, episodic memory, visuospatial abilities, language, and perceptual speed. Test scores were standardized into z-scores using the age-specific mean and standard deviation. Individuals who scored ≤ 1.5 SD below the age-specific mean in at least one cognitive domain were classified as cognitively impaired. **Those who scored above this threshold in all five domains were considered cognitively unimpaired.**

In addition to the neuropsychological assessment, participants underwent:

- a **comprehensive medical examination**, which included a detailed medical history and clinical evaluation, blood tests, review of inpatient and outpatient records, medical journals, and records from the Swedish National Patient Register to detect the presence of chronic diseases including dementia or other neurological conditions;
- a **full neurological examination**, which included the assessment of cognitive functions such as abstract thinking, orientation, problem solving, and general knowledge. Standard cognitive screening tools were administered, such as the Mini-Mental State Examination (MMSE), the Clock Drawing Test, forward and backward counting, and a short story recall task assessing frontal lobe function. A targeted examination for Parkinson’s disease and parkinsonism was also conducted.

Concerning the use of biomarkers to detect preclinical AD; in this study, AD blood biomarkers were used solely as exposure variables and were not involved in the classification of MCI, all-cause dementia, or Alzheimer’s disease. Diagnoses within the SNAC-K study are based on clinical assessments only – as detailed below – without the use of biological markers, neuroimaging, or neuropathological data. In this paper, our primary objective was to evaluate the association between blood biomarkers and clinically relevant outcomes such as MCI and dementia. As mentioned above in our response to Reviewer #1, we believe that these clinical diagnoses represent meaningful outcomes, particularly (but not only) in the context of large-scale, community-based studies. Moreover, in a population-based setting, dementia and MCI often result from heterogeneous and mixed pathologies, making it challenging to isolate “pure” AD—and even more so preclinical AD.

Reviewer’s comment 4: Was AD dementia defined solely based on clinical criteria? Were any CSF or PET biomarkers available to support the diagnoses?

Author’s reply: As mentioned above, SNAC-K is a population-based study, and the diagnosis of dementia and its subtypes is purely clinical, conducted without the use of biomarkers, neuroimaging, or neuropathology. This approach is **widely adopted in population-based cohorts**, where dementia is usually defined through standardized clinical criteria, as for example in previous studies conducted in SNAC-K and in the ARIC study (e.g., Lu et al., JAMA 2024; Grande et al., Nat Med 2025).

Clinical diagnoses in SNAC-K adhere to standard criteria. All-cause dementia is diagnosed according to the DSM-IV criteria and AD dementia according to the NINCDS-ADRDA criteria. The final diagnoses are made through a rigorous **three-step procedure**:

1. A preliminary diagnosis is made by a trained examining physician. The physicians conducting the individual face-to-face clinical examination have received ad-hoc training for this purpose.
2. A second physician, blind to the first diagnosis, conducts an independent review of the clinical records and made a second preliminary diagnosis.

3. In case of disagreement between the first and second diagnoses, a senior neurologist expert in cognitive assessment and dementia, and who are external to the data collection, makes the final diagnosis.

In addition, **for individuals who died between study waves without a diagnosis of dementia**, clinical records and the Swedish National Cause of Death Register were retrieved and reviewed by the same physicians involved in the diagnostic process. This approach minimizes the risk of underestimating dementia cases due to death occurring before a scheduled follow-up assessment.

We have now added further details about the diagnostic procedure in the Methods section of the revised manuscript.

We understand the potential concerns raised by the Reviewer regarding the lack of CSF and/or PET biomarkers to confirm an AD dementia diagnosis, which could indeed introduce a risk of misclassification. However, this reflects a common occurrence in community-based cohorts, where CSF or PET assessments are rarely available, thus supporting the generalizability and applicability of our findings to real-world scenarios. This limitation has been acknowledged and discussed as one of the main limitations of the present study:

“Some limitations should also be mentioned. First, dementia diagnosis was based solely on clinical assessment and did not incorporate neuroimaging or CSF biomarkers, introducing a potential risk of AD dementia misclassification. This challenge is inherent to community-based settings, where biological confirmation is rarely available and mixed etiologies are frequent. To enhance diagnostic accuracy, we adopted a standardized three-step procedure involving two trained physicians and a neurologist with expertise in dementia diagnosis. We also reviewed clinical charts and death registers to ensure the detection of dementia in deceased participants.”

Of interest, in an ongoing study (Mikulic I, Grande G, Rizzuto D et al., in preparation) we are linking data from a subgroup of SNAC-K participants with the Swedish Dementia Registry (SveDem; www.svedem.se), a national quality registry that collects detailed clinical information on dementia diagnoses established in both specialist and primary care settings. Preliminary results show a very high concordance between dementia diagnoses in SNAC-K and SveDem, not only for all-cause dementia but also for dementia subtypes. Thanks to these preliminary findings, we are further confident in the validity of our dementia diagnoses.

Nevertheless, acknowledging the Reviewer’s concern, we have expanded the Discussion section of the revised manuscript to better address the limitation related to the absence of biological confirmation of AD dementia cases.

Reviewer’s comment 5: The study reports a dropout rate of 6.2% (142 participants). How were these individuals handled in the analysis? Were any sensitivity analyses performed?

Author’s reply: The 142 participants who dropped out after the baseline assessment did not provide consent to link their data with the national patient or cause of death registers; therefore, we have no follow-up information regarding their cognitive outcomes. In our main analyses, these individuals were excluded, as they did not contribute any transition data, leaving an analytical population of 2148 participants. Compared to those with available follow-up data, participants who dropped out after the baseline assessment were younger (mean age difference: -7.52 years; 95% CI: -9.27, -5.76), more educated (university level: 47.2% vs. 35.4%, $p = 0.044$), and had fewer chronic diseases (mean difference: -1.00 diseases; 95% CI: -1.39, -0.60). This information has been added to the Methods section of the revised manuscript. To assess whether dropout could have biased our results, we performed **a logistic regression to identify baseline predictors of attrition**. As shown in the **Table** below, age was the only variable significantly associated with dropout. In our study, age was used as time scale in all the analyses.

Dropout	Odds ratio (95% CI)	p
Age	0.91 (0.88, 0.94)	0.000
Sex (F)	1.23 (0.85, 1.77)	0.272
Education		
Elementary	Ref.	
High school	0.64 (0.36, 1.14)	0.131
University or above	0.70 (0.39, 1.26)	0.235
Number of chronic diseases	0.96 (0.87, 1.06)	0.417

Prompted by the Reviewer's comment, and to further address the potential bias due to attrition, we repeated the analyses using **inverse probability weighting (IPW)**. The IPW weights were derived from logistic regression models including baseline age, sex, education, number of chronic diseases, and biomarker levels. These models estimated each participant's probability of remaining in the study, and the resulting weights were then applied in the multistate models. These additional analyses (shown in the **Tables** below) yielded results that were largely consistent with our main findings.

Table. Levels of blood biomarkers of Alzheimer's disease and hazard ratio (HR) of progression from normal cognition (NC) to mild cognitive impairment (MCI), reversion from MCI to NC, and progression from MCI to all-cause dementia, **applying inverse probability weighting (IPW)**.

	From NC to MCI	From MCI to NC	From MCI to all-cause dementia
	HR (95%CI)	HR (95%CI)	HR (95%CI)
Aβ-42/40			
Low vs High	0.96 (0.71, 1.30)	0.75 (0.50, 1.11)	1.30 (1.05, 1.60)
P-tau181			
High vs Low	0.92 (0.64, 1.32)	0.55 (0.34, 0.90)	1.37 (1.10, 1.70)
P-tau217			
High vs Low	1.07 (0.75, 1.53)	0.69 (0.41, 1.14)	1.75 (1.39, 2.19)
T-tau			
High vs Low	1.17 (0.85, 1.63)	1.10 (0.70, 1.72)	1.42 (1.15, 1.75)
NfL			
High vs Low	0.98 (0.65, 1.50)	0.62 (0.37, 1.05)	1.90 (1.49, 2.42)
GFAP			
High vs Low	0.92 (0.65, 1.30)	0.55 (0.35, 0.86)	1.49 (1.18, 1.87)

Hazard Ratios (HR) with 95% Confidence Intervals (CI) are derived from multistate Markov models, using age as time scale and adjusted for sex and education. Cut-offs: 0.057 for A β -42/40 ratio, 1.512 pg/mL for p-tau181, 0.134 pg/mL for p-tau217, 0.832 pg/mL for t-tau, 20.171 pg/mL for NfL and 142.515 pg/mL for GFAP.

Table. Levels of blood biomarkers of Alzheimer's disease and hazard ratio (HR) of progression from normal cognition (NC) to mild cognitive impairment (MCI), reversion from MCI to NC, and progression from MCI to AD dementia, **applying inverse probability weighting (IPW)**.

	From NC to MCI	From MCI to NC	From MCI to AD dementia
	HR (95%CI)	HR (95%CI)	HR (95%CI)
Aβ-42/40			
Low vs High	0.88 (0.66, 1.16)	0.66 (0.47, 0.92)	1.40 (1.09, 1.81)
P-tau181			
High vs Low	1.00 (0.73, 1.36)	0.59 (0.40, 0.87)	1.52 (1.18, 1.95)
P-tau217			
High vs Low	1.21 (0.89, 1.65)	0.75 (0.50, 1.11)	2.11 (1.62, 2.76)
T-tau			
High vs Low	1.09 (0.82, 1.44)	0.98 (0.70, 1.38)	1.44 (1.12, 1.84)
NfL			
High vs Low	1.20 (0.87, 1.66)	0.72 (0.49, 1.06)	2.36 (1.79, 3.11)
GFAP			
High vs Low	1.13 (0.83, 1.53)	0.67 (0.47, 0.97)	1.90 (1.44, 2.50)

Hazard Ratios (HR) with 95% Confidence Intervals (CI) are derived from multistate Markov models, using age as time scale and adjusted for sex and education. Cut-offs: 0.057 for A β -42/40 ratio, 1.512 pg/mL for p-tau181, 0.134 pg/mL for p-tau217, 0.832 pg/mL for t-tau, 20.171 pg/mL for NfL and 142.515 pg/mL for GFAP.

These sensitivity analyses have now been added to the revised manuscript as **Supplementary Tables S8 and S9**.

Reviewer's comment 6. While the study clearly shows biomarker associations with cognitive progression at the group level, the authors should address whether these findings are translatable to the individual level. This is critical to support the conclusion that "individuals with mild cognitive deficits" are an appropriate target for community-based biomarker testing.

Authors' reply: We thank the Reviewer for raising this important point regarding the translation of group-level biomarker associations to individual-level predictions. Our study was not designed to provide individual risk predictions, and we agree with the Reviewer that our findings apply at the group level. Statements implying individual-level utility have been removed (e.g., "These results highlight individuals with mild cognitive deficits – rather than cognitively unimpaired individuals – as a more suitable target for biomarker testing in the community." in the Conclusions).

In our study population, 381 participants had MCI at baseline and 286 developed MCI during follow-up. Among those with MCI at baseline, 146 had high levels of p-tau217, 161 of NfL and 164 of GFAP. While these numbers were sufficient to establish robust group-level associations, they were too small to allow reliable individual-level prediction analyses. The relatively limited number of dementia cases further constrained the possibility of deriving individual prediction metrics. We agree that translation to the individual level is a crucial next step. Nonetheless, our study provides an important foundation, as it demonstrates that AD blood biomarkers (and their combinations) are associated with the risk of progression from MCI to all-cause and AD dementia in community-dwelling older adults.

The revised Discussion and Conclusions emphasize that our results indicate that AD blood biomarkers stratify group risk of progression from MCI to dementia and highlight the need for larger dedicated studies to develop and validate individual-level tools for dementia prediction.

In the Conclusions:

“In this large community-based study, elevated levels of several AD blood biomarkers – particularly p-tau217, NfL, and GFAP – were associated with the progression from MCI to all-cause and AD dementia and with decreased MCI reversion. Notably, no association emerged between biomarker levels and the development of MCI in cognitively unimpaired participants. These findings demonstrate robust group-level associations and suggest that AD blood biomarkers may help stratify the risk of progression to dementia at the MCI stage in the community. Future studies including larger cohorts of individuals with mild cognitive deficits should explore the integration of clinical and biomarker data to improve individual-level dementia prediction.”

Reviewer #3:

Reviewer’s comment: Thank you for the opportunity to review the above manuscript and I commend the authors for their detailed exploration of the relationship between BBM of AD and progression in dementia free adults.

I feel that this would be a much more impactful manuscript if authors could change from analysing z-scores of biomarkers to predict progression in CIND to biomarker "positivity" (particularly for p-tau217) in MCI - this in my mind would have a significant impact on the applicability of these findings to a broader audience and in particular is an incredibly useful clinical question. This would require: (i) cleaning up definition of CIND/MCI to reflect individuals who are functionally intact (using 1.5SD below the mean as cut off is acceptable in my view, this small step would 'complete' the definition) and (ii) defining cut off values for p-tau217, p-tau181 and ratios at least to give us HR for conversion in MCI with positive blood biomarkers.

Authors’ reply: We thank the Reviewer for the feedback. Prompted by the Reviewer’s suggestions, we have revised the operational definition of cognitive impairment used in our study. Specifically, we replaced the construct “cognitive impairment, no dementia (CIND)” — which was based solely on lower cognitive performance exceeding the expected level for the individual’s age — with “mild cognitive impairment (MCI)”, which additionally incorporates information on the individual’s functional ability. We have applied this new operational definition throughout the revised manuscript and rerun all the analyses accordingly. Additionally, the new analyses using dichotomized biomarkers, using previously defined cut-offs from the same cohort (Grande et al., Nat Med 2025), are reported in the revised **Table 2** (for all-cause dementia) and **Supplementary Table S1** (for AD dementia). A point-by-point response to the Reviewer’s comments follows below.

Reviewer’s comment: Notable points:

- The study has over two thousand participants, from the SNAC-K cohort. As far as I'm aware there is limited data in these types of cohorts using BBMs, so this is commendable.
- Study is commendable for 16 year follow up. Many studies have only examined short term
- Also commendable to look at a panel of BBMs, not just leading BBM.
- Study setting and recruitment well described
- The description of biomarker analysis using Simoa is excellent; I cannot comment specifically on the spline modelling as I haven't carried out this analysis myself, but the z-scoring makes sense and description of the modelling is accurate and appropriate.
- I welcome the sex specific results, mirroring recent data that demonstrates higher levels of AD biomarkers in females.

Authors’ reply: We sincerely thank the Reviewer for the positive comments. We are pleased that the Reviewer appreciated the strengths of our study

Points for reflection/discussion

Reviewer's comment 1: - CIND is not necessarily a well-defined term? I would suggest using MCI as this is better understood, I would refer authors to the Manchester consensus on MCI, published in recent years in age and ageing by ross dunne and colleagues.

- From the methods section, it appears that CIND is 1.5 SD below norms on neuropsych function - this is the exact definition of MCI. <https://pubmed.ncbi.nlm.nih.gov/33197937/>

- Does SNAC-K have an ADL questionnaire - this should be added to determine if those with MCI/CIND are functionally intact (may lead to excluding a tiny minority of participants, but clinically this would make a lot more sense).

Author's reply: We thank the Reviewer for these thoughtful considerations. We agree that, while CIND is a construct commonly used in epidemiological studies, MCI is a more widely recognized and clinically meaningful construct. Data on participants' functional independence are available in SNAC-K, as we collected information on both basic (ADL) and instrumental (IADL) activities of daily living. These data have now been used to refine the classification of cognitive status together with functional dependence. Prompted by the Reviewer's valuable suggestion, we have revised both the analyses and the manuscript to adopt the operationalization of "MCI" instead of "CIND".

Following the Manchester consensus (Dunne et al., Age and Ageing, 2021), which the Reviewer kindly referred to, we now operationalize MCI as:

- scoring ≤ 1.5 SD below the age-specific mean in at least one cognitive domain
- with largely preserved daily functioning — defined as having no more than one impaired instrumental activity of daily living (IADL) and preserved basic activities of daily living (ADL) — in the absence of a dementia diagnosis.

This definition was applied consistently across baseline and follow-up assessments.

Importantly, the main findings of our study remain largely unchanged when using this MCI-based definition, and the overall interpretation of the results is consistent with our original analyses. Specifically:

- Individuals with higher p-tau181, p-tau217, t-tau, NfL, and GFAP experienced a faster progression from MCI to all-cause and AD dementia compared to those with lower biomarker levels.
- These associations were even stronger when multiple biomarkers were considered together.
- A low A β 42/40 ratio was also associated with the progression from MCI to all-cause and AD dementia.
- High NfL and GFAP were associated with a less likely reversion from MCI to normal cognition.
- No association emerged between any of the examined biomarkers and the development of MCI among cognitively unimpaired participants.

Reviewer's comment 2: Were all individuals examined for dementia at all study visits? This should be made clear.

Author's reply: Yes, all participants were systematically assessed for dementia **at baseline and at each follow-up wave** using a standardized and rigorous **three-step procedure**:

1. A first preliminary diagnosis was made by the examining physician, who conducted the in-person clinical assessment. These physicians were trained specifically in dementia diagnosis and had substantial experience in this area.
2. A second physician, blinded to the initial assessment, independently reviewed the clinical documentation and made a second preliminary diagnosis.
3. In the case of disagreement between the two, a senior neurologist with expertise in cognitive disorders—external to the data collection process—reviewed the case and issued a final diagnosis.

All-cause dementia diagnoses were based on DSM-IV criteria, while AD dementia was diagnosed according to the NINCDS-ADRDA criteria.

In addition, **for individuals who died between study waves without a diagnosis of dementia**, clinical records and the Swedish National Cause of Death Register were retrieved and reviewed by the same physicians involved in the diagnostic process. This approach minimizes the risk of underestimating dementia cases due to death occurring before a scheduled follow-up assessment.

Following the Reviewer's suggestion, we have now clarified the methodology used for dementia diagnosis in the Methods section of the revised manuscript:

"At baseline and at each follow-up assessment, all-cause and AD dementia were diagnosed following the Diagnostic and Statistical Manual of Mental Disorders, 4th Edition, (DSM-IV) criteria⁵¹ and the NINCDS-ADRDA criteria⁵². The diagnosis followed a three-step procedure..."

Reviewer's comment: - Did authors look at p-tau217/amyloid ratios - given recent FDA approval of lumipulse for this ratio

Author's reply: We thank the Reviewer for this suggestion. Prompted by this comment, we computed the p-tau217/A β 42 ratio and repeated the analyses (shown in the **Figures** and **Tables** below).

Figure. Continuous values of blood p-tau217/A β 42 ratio and hazard ratio (HR) of progression from normal cognition (NC) to mild cognitive impairment (MCI), reversion from MCI to NC, and progression from MCI to all-cause dementia.

Figure. Continuous values of blood p-tau217/Aβ42 ratio and hazard ratio (HR) of progression from normal cognition (NC) to mild cognitive impairment (MCI), reversion from MCI to NC, and progression from MCI to AD dementia.

Table. Levels of blood p-tau217/Aβ42 ratio and hazard ratio (HR) of progression from normal cognition (NC) to mild cognitive impairment (MCI), reversion from MCI to NC, and progression from MCI to all-cause dementia.

	From NC to MCI		From MCI to NC		From MCI to all-cause dementia	
	Basic HR (95%CI)	Fully adj HR (95%CI)	Basic HR (95%CI)	Fully adj HR (95%CI)	Basic HR (95%CI)	Fully adj HR (95%CI)
P-tau217 /Aβ42						
High vs	1.09	1.08	0.75	0.79	1.50	1.40
Low	(0.79, 1.51)	(0.76, 1.55)	(0.47, 1.20)	(0.48, 1.33)	(1.20, 1.87)	(1.12, 1.76)

Cut-off for p-tau217/Aβ42: 0.019

Table. Levels of blood p-tau217/Aβ42 ratio and hazard ratio (HR) of progression from normal cognition (NC) to mild cognitive impairment (MCI), reversion from MCI to NC, and progression from MCI to AD dementia.

	From NC to MCI		From MCI to NC		From MCI to AD dementia	
	Basic HR (95%CI)	Fully adj HR (95%CI)	Basic HR (95%CI)	Fully adj HR (95%CI)	Basic HR (95%CI)	Fully adj HR (95%CI)
P-tau217 /Aβ42						
High vs	1.15	1.12	0.75	0.75	1.72	1.59
Low	(0.86, 1.54)	(0.83, 1.51)	(0.51, 1.09)	(0.51, 1.11)	(1.33, 2.23)	(1.23, 2.06)

Cut-off for p-tau217/Aβ42: 0.019

According to these additional analyses, participants with higher p-tau217/Aβ42 ratio values had an increased hazard of progression from MCI to both all-cause dementia (HR 1.40, 95% CI 1.12, 1.76, fully

adjusted model) and AD dementia (HR 1.59, 95% CI 1.23, 2.06), compared to those with lower levels of the ratio.

However, these associations were slightly weaker than those observed for p-tau217 alone (HR 1.64, 95% CI 1.29, 2.08 for all-cause dementia and HR 1.87, 95% CI 1.43, 2.46 for AD dementia). This may be attributable to known limitations in the measurement of A β 42 in blood, as discussed in the manuscript. Specifically, plasma A β concentrations are substantially lower than those in CSF—up to tenfold—and are affected by peripheral amyloid production (Jack et al., *Alzheimers Dement* 2024; Roher et al., *Alzheimers Dement* 2010). Previous studies have shown that blood A β levels, especially when measured using immunoassays, correlate less strongly with brain amyloid deposition than CSF measurements (Janelidze et al., *JAMA Neurology* 2021).

Considering these limitations, and the non-superiority of the p-tau217/A β 42 ratio compared to p-tau217 alone in our analyses, we decided not to include this marker in the set of biomarkers presented in the manuscript. Additionally, we used a different assay than Lumipulse for the quantification of A β 42 and p-tau217, which may complicate direct comparisons with studies employing that platform.

Reviewer's comment: Individuals are characterised with z-score of biomarker. This is definitely an acceptable way to perform this analysis. However, what would really increase the clinical applicability of the findings is to get a HR for biomarker positivity. This particularly applies to p-tau217. If the authors have established cut-offs in the same lab on the same machine (for instance with participants with paired blood/CSF available - or from an existing biobank that could be run), define positivity cut-offs and use these to predict conversion.

Author's reply: We appreciate the Reviewer's comment and fully agree that dichotomizing biomarkers using clinically meaningful cut-offs can enhance the clinical applicability of our findings. Unfortunately, no established cut-offs derived from paired blood/CSF measurements are currently available, nor were such reference values generated using the same lab and Simoa platform. To address this, **we derived cut-offs empirically within our cohort**. Specifically, we randomly split the study population into training (80%) and testing (20%) datasets. In the training dataset, we applied a non-parametric bootstrapping approach (5,000 resamples) to identify the optimal cut-off for each biomarker to predict 10-year all-cause dementia, maximizing Youden's Index. The predictive performance of these cut-offs was then evaluated in the testing dataset. This approach, while internal, allowed us to estimate robust and data-driven thresholds for biomarker positivity. This methodology was previously applied in a published study from our group (Grande et al., *Nat Med* 2025).

We have now clarified this methodology in the revised Methods section and presented hazard ratios based on these biomarker-positive categories in the revised **Table 2** (for all-cause dementia) and **Supplementary Table S1** (for AD dementia).

Point by point response to comments from the Reviewers

Reviewer #2: The authors successfully addressed all my comments. I believe that the manuscript can be accepted in its current form.

Author's reply: We appreciate that the Reviewer is satisfied with the revised version of the manuscript. We sincerely thank the Reviewer again for the comments and suggestions, which have improved the quality and clarity of our work.

Reviewer #3: I am satisfied that all my concerns have been addressed in the revision - the authors should be commended on their work

Author's reply: We are pleased that the Reviewer is satisfied with our revisions. We sincerely thank the Reviewer for the constructive feedback, which helped us strengthen the manuscript and improve its quality.

Reviewer #4:

- Reviewer #1
 - 1) Novelty vs. prior SNAC-K/Nat Med (Grande et al., Nat Med 2025) paper: In the rebuttal and revision the authors (i) clarify that the prior paper modeled incident dementia only, whereas this study uses multistate Markov models to follow all transitions (NC→MCI, MCI→NC, MCI→dementia) over up to 16 years, and (ii) explicitly re-frame the contribution as showing where blood biomarkers are most informative (i.e., at the MCI stage) in a community cohort. They expanded Intro/Discussion accordingly. This is a reasonable and, in my view, sufficient differentiation; while the novelty is incremental, it is real.
 - 2) Use of CIND vs MCI: The authors re-ran the set of analysis using a standard MCI operationalization that incorporates ADL/IADL (≤ 1 impaired IADL; preserved BADLs) and show the main conclusions are unchanged. The Methods now detail this change. It was adequately addressed.
 - 3) Clarity of the biomarker-combination analysis: The authors explain they did model four groups for each pair (low/low; high NfL only; high p-tau217 only; high/high), removed the confusing panel, and provided the full estimates in Supplementary Tables. It was adequately addressed.
 - 4) Clinical (not biomarker) dementia ascertainment: The revision now describes the three-step clinical adjudication and openly acknowledges the lack of CSF/PET as a limitation. This is transparent and appropriate for a community cohort.

Author's reply: We are glad that the Reviewer considers our responses to Reviewer #1 appropriate and satisfactory.

- Reviewer #3
 - 1) Analyze positivity in MCI (esp. p-tau217) rather than z-scores in CIND: They switched from CIND to MCI and added analyses using dichotomized biomarker cutoffs (including p-tau217) previously derived within SNAC-K (Youden's index on a training set with hold-out testing), reporting HRs for progression. This directly meets the request. (E.g., high p-tau217 and high NfL each predict faster MCI→AD dementia; effects are strongest when both are elevated.)
 - 2) Clarify MCI definition and functional intactness: They now require ≤ 1.5 SD impairment in ≥ 1 domain and preserved function (ADL/IADL) per Manchester consensus; applied at baseline and follow-ups. It was adequately addressed.
 - 3) Consider the p-tau217/A β 42 ratio: In response, they computed p-tau217/A β 42 and repeated the transition analyses (tables/figures shown in rebuttal). This was responsive.

4) Clinical diagnosis details & external validity: They added detail on the three-step adjudication, noted the lack of CSF/PET, and flagged ongoing registry linkage, suggesting high concordance of dementia diagnoses—appropriately framed as supportive but preliminary.

5) Avoid implying individual-level prediction: They explicitly removed individual-prediction language, restricting claims to group-level risk stratification. This was responsive.

Author's reply: We are pleased that the Reviewer found our revisions and additional analyses responsive to all previous comments.

- My own thoughts

1) This is a strong community-based analysis with long follow-up, but the sample's diversity and external validity are limited: SNAC-K is a single urban Swedish cohort with high educational attainment (35.4% university) and a majority of women (61.5%); race/ethnicity is not reported in the baseline table, and the study setting is a relatively homogeneous, high-resource health system. As a result, performance and cut-points for blood biomarkers derived internally via an 80/20 split on this same cohort and a specific Simoa/ALZpath platform may not transfer to more diverse populations by race/ethnicity, socioeconomic status, rurality, or to different laboratory platforms without recalibration.

Author's reply: We thank the reviewer for highlighting this important point. We have now explicitly acknowledged this limitation in the revised manuscript:

Third, our results based on cut-offs derived within the SNAC-K cohort, may not fully extend to more diverse populations or alternative laboratory platforms. Future studies are needed to validate these findings in independent and more diverse cohorts.

This clarifies that, although SNAC-K is a large, population-based cohort with extensive follow-up, which provides a robust foundation for our analyses, further studies are needed to confirm the applicability of these findings across populations with different demographic, socioeconomic, and geographic characteristics, as well as on alternative assay platforms.

2) Methodologically, the multistate Markov framework is a strength, but several assumptions and design features temper robustness: (i) transitions are observed at visits, with the model assuming no direct normal to dementia progression (unobserved MCI is imputed), which can misclassify short prodromal intervals; (ii) biomarkers were measured once at baseline, so dynamic changes or regression dilution were not captured; and (iii) dementia diagnoses rely on clinical criteria without CSF/PET confirmation in a setting with common mixed pathologies, inviting outcome misclassification despite a careful three-step adjudication and register linkage.

Author's reply: We thank the Reviewer for these methodological considerations.

- (i) Regarding the first point, the specific multistate Markov model we used was chosen because it allows us to account for interval-censoring, meaning that transitions are observed only at study visits rather than continuously (*Jackson et al. Multi-state Survival Models for Interval-censored Data. Biometrics 2018; Jackson et al. Multi-State Models for Panel Data: The msm Package for R. J. Stat. Softw. 2011*). This feature helps mitigate, though does not eliminate, the issues raised by the reviewer related to unobserved short prodromal intervals.
- (ii) Biomarker measurements were available only at baseline, and we have therefore noted the lack of longitudinal biomarker data in the limitations section: *Finally, the availability of biomarkers only at baseline did not allow us to assess the associations between changes in biomarker levels and progression across stages of cognitive decline.*
- (iii) We acknowledge that the lack of CSF/PET confirmation may lead to misclassification specifically for AD dementia, while it is unlikely to affect all-cause dementia. This limitation

has already been noted and discussed in the manuscript. We also emphasize that clinical diagnosis remains the standard in community-based settings, where biological confirmation is rarely available and mixed pathologies are common.

3) Selection also matters: 833 otherwise eligible participants lacked baseline biomarkers and were older, less educated, and sicker, and 6.2% dropped out after baseline; both patterns could bias associations (likely toward underestimation per authors' discussion), although IPW analyses were performed and mostly aligned with the main results.

Author's reply: As noted by the Reviewer, in the manuscript we present a comparison between participants with available versus missing biomarker data, as well as those with available versus missing follow-up data, and we discuss the potential implications for our results.

4) Finally, while findings for MCI→dementia were consistent (strongest for p-tau217, NfL, GFAP), some effects attenuate in fully adjusted and sensitivity models (e.g., p-tau181 for MCI reversion, and certain associations when baseline MCI is excluded), underscoring that individual-level prognostication is not yet supported and that broader, multi-site validation with standardized assays and pre-specified (potentially age/sex-specific) thresholds is still needed before clinical deployment in heterogeneous populations.

Author's reply: As pointed out by the Reviewer, in the Discussion and Conclusions we emphasize that, although our results are promising and robust at the group level, further work is still needed before AD blood biomarkers can be applied clinically for individual-level dementia prediction in the general population.